# Rational tuning of temperature sensitivity of the TRPM8 channel

Lizhen Xu [ID][1,2,3,10], Xiao Liang [ID][4,5,10], Yunfei Wang [ID][6,10], Han Wen[7,10], Wenxuan Zhen[1,2,3], Zhangzhi Xue[4,5], Fangfei Zhang[4,5], Xiao Yi[4,5], Xiaoying Chen[1,2,3], Lidan Hu[8], Bei Li[3,9], Bing Zhang[3,9], Zhenfeng Deng [ID][7], Wei Yang [ID][1], Shilong Yang [ID][6✉], Tiannan Guo [ID][4,5✉], Yi Zhu [ID][4,5✉] & Fan Yang [ID][1,2,3✉]

## Abstract

Detecting temperature is crucial for the survival of living organisms. Although the temperature sensitive Transient Receptor Potential Melastatin 8 (TRPM8) channel has been identified as the prototypical cold sensor, the mechanisms by which it detects temperature remain elusive. In this study, we first identify groups of clustered residues that undergo conformational rearrangements between buried and exposed states during cold activation by hydroxyl radical footprinting-mass spectrometry (HRF-MS). By systematically perturbing water-protein interactions at these residues with point mutations that change side chain hydrophobicity (SCH), we achieve rational tuning of temperature sensitivity in this channel. Specifically, mutations with the clearest impacts on TRPM8 cold sensitivity are clustered in the MHR1-3 domains, where the protein of isolated MHR1-3 domains also exhibits clear conformational rearrangements in response to cold. Guided by this mechanism, we rationally edit the *Trpm8* gene in mice, introducing a single point mutation to render them insensitive to coldness.

**Keywords** TRPM8 Channel; Temperature Sensing; Water–Protein Interactions; Conformational Rearrangements
**Subject Categories** Membranes & Trafficking; Neuroscience

## Introduction

Temperature sensation is vital for survival across species. Members of the transient receptor potential (TRP) family, particularly TRPV1 (Caterina et al, 1997) and TRPM8 (McKemy et al, 2002; Peier et al, 2002), are well-established molecular thermosensors that are highly sensitive to heat and cold, respectively. The TRPM8 channel is activated by low temperatures (<28 °C), and the

enthalpic (ΔH) and entropic (ΔS) changes associated with its temperature gating are considerably larger than those observed in typical proteins (Julius, 2013; Zheng, 2013). Moreover, the TRPM8 channel functions as a temperature detector in vivo, so when it is genetically knocked out (Caterina et al, 2000; Dhaka et al, 2007) or pharmacologically inhibited (Moran, 2018), the temperature-sensing ability in animals is significantly disrupted. Despite the critical physiological roles of the TRPM8 channel, the mechanisms by which it detects temperature and whether a common temperature-sensing mechanism is employed by the TRPM8 channel remain largely elusive.

The hypothesis that changes in the buried/exposed state (and consequently water–protein interactions) of amino acids serves as a mechanism for temperature sensing (Figs. 1A and EV1A) (Clapham and Miller, 2011a; Yeh et al, 2023) was preliminarily tested in our study of TRPM8 cold activation (Yang et al, 2020) and supported by findings from the Shaker Kv channel (Chowdhury et al, 2014). In the cold-sensitive TRPM8 channel, decreasing temperature reduces water dynamics, forming an ordered hydration shell around exposed residues. Hydrophobic sidechains stabilize within this shell, yielding positive ΔCp and more negative ΔH and ΔS, thereby enhancing cold sensitivity. Conversely, exposure of polar or charged residues disrupts the hydration shell and weakens cold sensitivity, whereas increased polarity of buried residues has the opposite effect (Figs. 1A and EV1A)(Makhatadze et al, 1990; Makhatadze and Privalov, 1993; Privalov and Makhatadze, 1990). A theoretical analysis further suggests that the temperature-sensing apparatus in TRP channels could be distributed on a single-residue basis, where changes in the buried/exposed state of 10–20 residues per subunit would be enough to drive temperature activation (Clapham and Miller, 2011b).

Previously, we employed the fluorescent unnatural amino acid 3-(6-acetylnaphthalen-2-ylamino)-2-aminopropanoic acid (ANAP) to preliminarily test the water–protein interaction hypothesis (Yang et al, 2020), but the approach was low-throughput and restricted to only a few accessible positions. HRF-MS offers advantages such as high throughput and minimal perturbation to sidechains, and has

[1]Kidney Disease Center of the First Affiliated Hospital and Department of Biophysics, Zhejiang University School of Medicine, Hangzhou, Zhejiang Province, China. [2]Liangzhu Laboratory, Zhejiang University Medical Center, Hangzhou, Zhejiang 311121, China. [3]Alibaba-Zhejiang University Joint Research Center of Future Digital Healthcare, Hangzhou, China. [4]Westlake Laboratory of Life Sciences and Biomedicine, Key Laboratory of Structural Biology of Zhejiang Province, School of Life Sciences, Westlake University, Hangzhou, Zhejiang Province, China. [5]Institute of Basic Medical Sciences, Westlake Institute for Advanced Study, Hangzhou, Zhejiang Province, China. [6]College of Wildlife and Protected Area, Northeast Forestry University, 150040 Harbin, Heilongjiang Province, China. [7]DP Technology, Beijing, China. [8]The Children's Hospital, Zhejiang University School of Medicine, National Clinical Research Center for Child Health, Hangzhou, Zhejiang Province, China. [9]Alibaba Group, Hangzhou, China. [10]These authors contributed equally: Lizhen Xu, Xiao Liang, Yunfei Wang, Han Wen. ✉E-mail: syang2020@nefu.edu.cn; guotiannan@westlake.edu.cn; zhuyi@westlake.edu.cn; fanyanga@zju.edu.cn

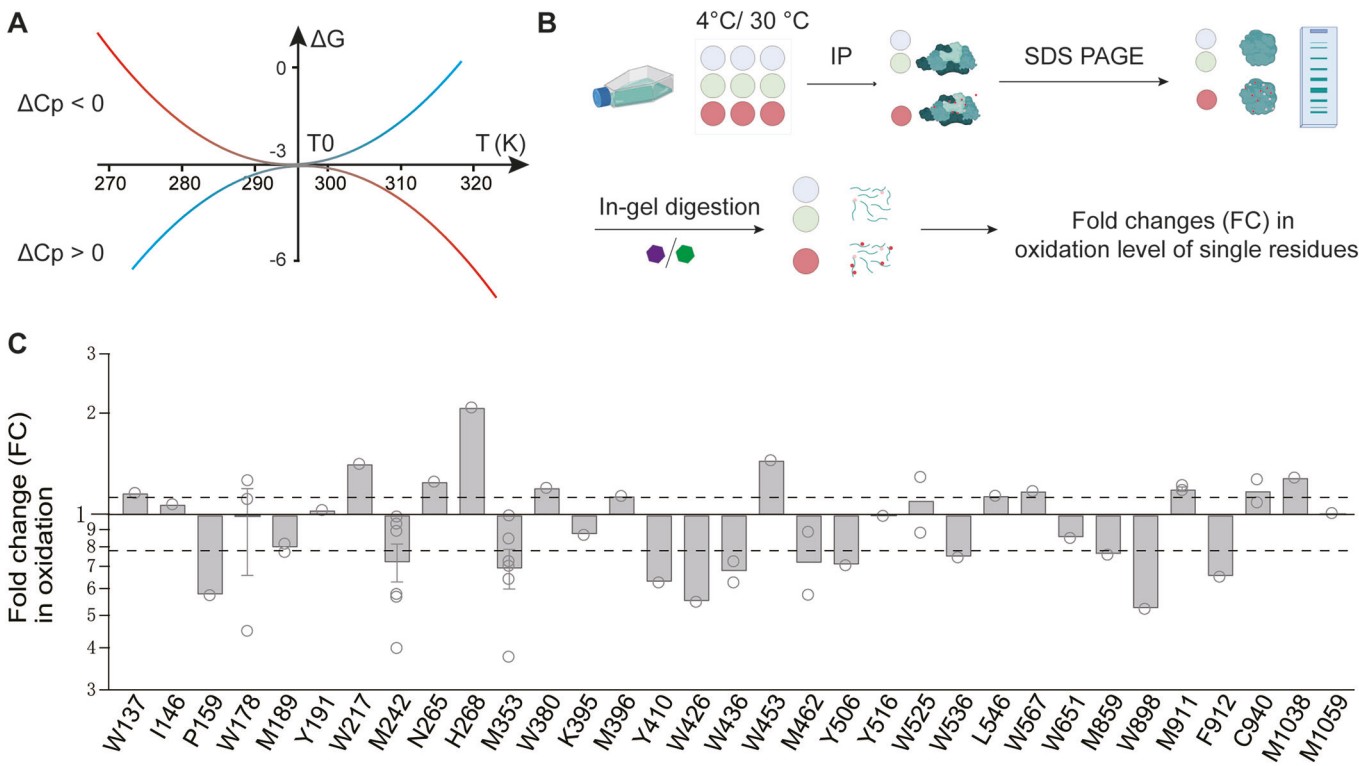

**Figure 1. Detecting residues with temperature-induced buried/exposed changes in TRPM8.**

(A) Simulation of the temperature dependence of ΔG during TRP channel activation with different ΔCp values. A positive ΔCp generates a convex curve, while a negative ΔCp produces a concave curve. The heat-sensing and cold-sensing regimes of the curves are indicated in red and blue, respectively. The ΔG-temperature profiles were simulated using the equation $\Delta G = \Delta H_0 + \Delta Cp(T - T_0) - \Delta S_0 T - T\Delta Cp \ln(T/T_0)$, where the values of positive ΔCp, negative ΔCp, $T_0$, $\Delta H_0$ and $\Delta S_0$ were set to be 3.00 kcal/(mol·K), −3.00 kcal/(mol·K), 298.15 K (25 °C), −3.00 kcal/mol and –0.01 kcal/(mol·K), respectively. (B) The workflow for TRPM8 oxidation site determination. (C) Oxidation experiments were performed at least three times with biological replicates. Bar graph for the FCs showing significant changes in oxidation levels of TRPM8 peptides. FC >1.2 or FC <0.83 under 4 and 30 °C. FC fold change. Solid line, 1.0; Dash line, 1.2 and 0.83. All detected TRPM8 residues with FCs are shown. For residues detected more than three times in the oxidation experiments, data were presented as mean ± s.e.m. Source data are available online for this figure.

become a powerful technique for probing higher-order protein structures and conformational dynamics (Gupta et al, 2010; Li et al, 2018; Liu et al, 2020a; Liu et al, 2020b; Sharp et al, 2004; Xie et al, 2017). It has been applied to systems such as the OmpF channel (Zhu et al, 2009a) and EGFR (Zhu et al, 2017), as well as to studies of protein-protein interactions (Espino et al, 2015; Kaur et al, 2020; Shortt et al, 2023; Sun et al, 2025) and in vivo models (Espino and Jones, 2019). Building on these advances, we performed HRF-MS on the TRPM8 channel to detect conformational changes during temperature activation (Figs. 1B and EV1B,C), followed by patch-clamp validation in mutants with altered SCH.

## Results and discussion

### Detecting residues with temperature-induced buried/exposed changes

To comprehensively identify residues that underwent buried/exposed changes during the cold activation of TRPM8, we performed an optimized HRF-MS analysis of the TRPM8 channel protein expressed in live cells at either 30 or 4 °C, where TRPM8 remained closed or was activated by cold, respectively (Figs.

1B and EV1B–E and Methods). In brief, we transiently over-expressed the TRPM8 channel proteins in HEK293 cells. In the living and TRPM8-expressing cells, we generated hydroxyl radicals through an in situ Fenton reaction involving hydrogen peroxide and Fe(II)-bound EDTA (Zhu et al, 2009a), which could easily diffuse across the cell membrane to modify proteins with exposed sidechains. Modern MS techniques enable the precise detection of peptide fragments with various modifications, including the oxidation of even a singleresidue (Aebersold and Mann, 2016). Two controls were set as no treatment (Condition C) and treatment with Fe(II)-bound EDTA, respectively. In this way, the sidechains in TRPM8 were oxidized within their native membrane environment before cell lysis, so that the fidelity of information on buried/exposed states of TRPM8 was warranted. In addition, we performed patch-clamp recordings on the TRPM8-expressing cells after Fenton oxidation, where we observed that the channels remained functional (Fig. EV2A,B).

We analyzed our HRF-MS results and found that the data were composed of two parts: the generation of the TRPM8 spectral library and the quantification of oxidation. For library generation, TRPM8 and its binding partners were firstly detected through MS acquisition in data-dependent acquisition (DDA) mode and processed into a complex library (Zhu et al, 2020a), which was

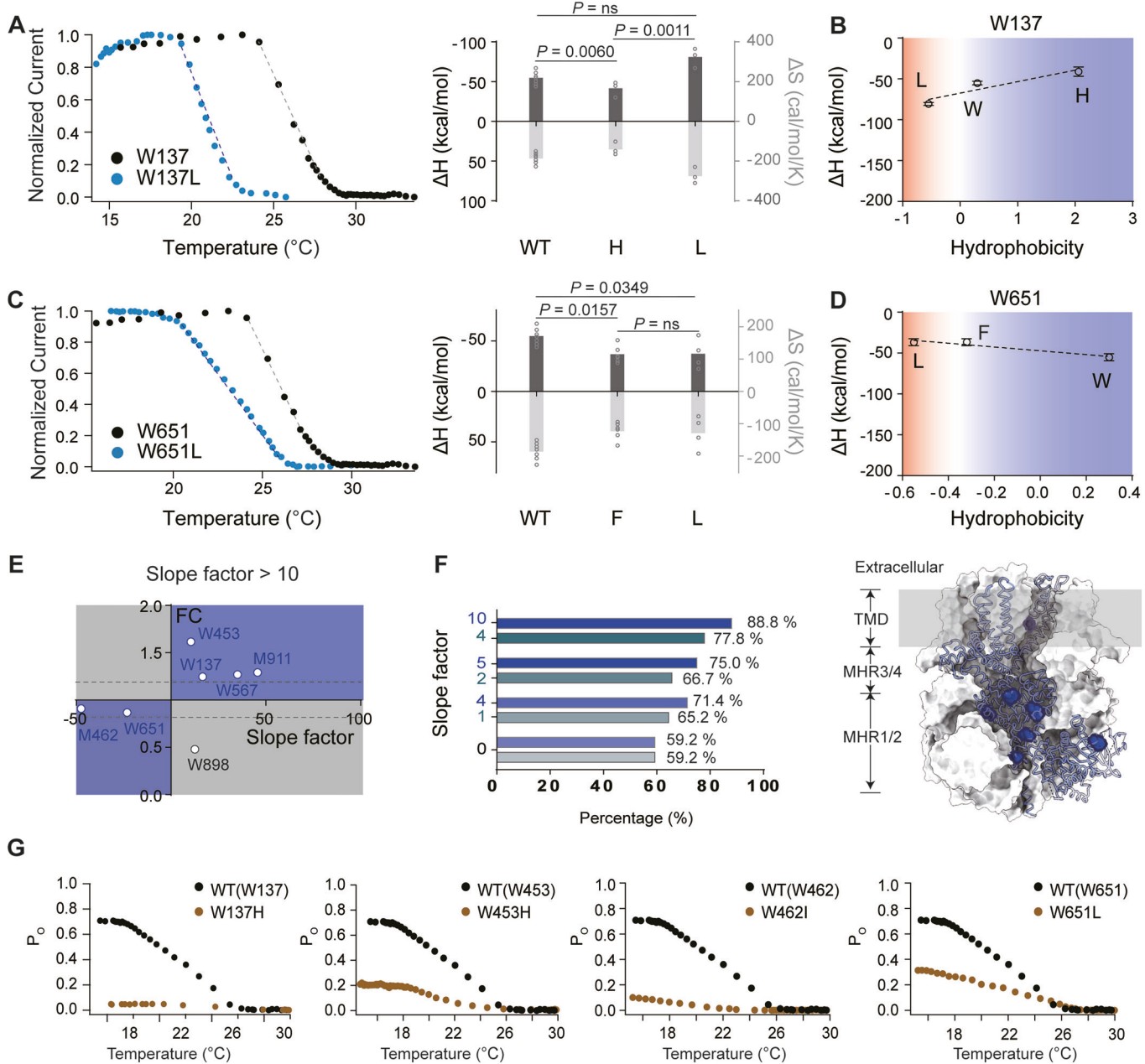

thereafter transferred as a database to downscale the search space. The TRPM8 peptides were acquired in DDA MS mode and were identified with the downscaled database (false discovery rate ≤0.01) to profile and generate a TRPM8-specific spectral library (Fig. EV1B). A total of 83.6% of the primary amino acid sequence of TRPM8 was covered by the library, which included 1777 peptides and 810 oxidation sites (Fig. EV1B). An example of the fragmentation spectrum of a peptide from the library is presented in Fig. EV1D.

For the quantification of oxidation efficiency, we acquired TRPM8 peptides using data-independent acquisition (DIA) MS mode at either 30 or 4 °C. The peptides were matched against the spectral library and quantified based on the summed intensities at MS1 and MS2 (Fig. EV1C). By conducting mass spectrometry (MS)

profiling, we assessed the oxidation levels of TRPM8 peptides based on their relative oxidation efficiency. For instance, we measured the extracted ion chromatogram (XIC) groups of the matched peptide [415]AFSTNEQDKDNWNGQLK[431] (Fig. EV1E), which were used to calculate the relative oxidation efficiency.

We further compared the fold change (FC) in relative oxidation levels measured at 4 and 30 °C. If the FC value of a peptide or residue is larger than 1.20 or smaller than 0.83 (Zhu et al, 2009a; Zhu et al, 2011), we regarded this peptide became significantly more exposed or buried at 4 °C, respectively (Fig. 1C; Table EV1). Only the peptides containing a single oxidized residue were considered. In this way, we identified 33 individual residues that exhibited cold-induced buried/exposed changes, as shown in HRF-MS (Table EV1; Datasets EV1 and 2).

**Figure 2. Clustering of TRPM8 residues with correlated changes in SCH and cold sensitivity in the MHR1-3 domains.**

(A) Left: representative temperature-driven activation of wild-type TRPM8 and its W137L mutant. Right: Measured ΔH (black bars, left axis) and ΔS values (light gray bars, right axis) of TRPM8 and the channel mutants ($n = 3$–7 biological replicates; statistical significance was determined using one-way ANOVA and Tukey's multiple comparisons test for histograms. Data were presented as mean ± s.e.m.). (B) Correlation between SCH and ΔH values. For TRPM8, the W137L residue with more exposed sidechains in cold activation. The hydrophobicity scale of SCH was determined by Hessa et al. (C) Left: representative temperature-driven activation of wild-type TRPM8 and its W651L mutant. Middle: Van't Hoff plots for the cold-activated TRPM8 currents shown in left. Right: Measured ΔH (black bars, left axis) and ΔS values (light gray bars, right axis) of TRPM8 and the channel mutants ($n = 3$–7 biological replicates; statistical significance was determined using one-way ANOVA and Tukey's multiple comparisons test for histograms. Data are presented as mean ± s.e.m.). (D) Correlation between SCH and ΔH values. For TRPM8 W651L residue with more buried sidechains in cold activation. The hydrophobicity scale of SCH was determined by Hessa et al. (E) Correlation between SCH and ΔH values for TRPM8 residues with changes in FC values during cold activation. The hydrophobicity scale of SCH was determined by Hessa et al. The slope factors greater than 10 were plotted against the FC values of the corresponding sites. The first and third quadrants, where the sites followed the predictions from the water–protein interaction hypothesis of cold sensing, were shaded in blue. The second and fourth quadrants, where the sites did not follow the predictions from the water–protein interaction hypothesis of cold sensing, were shaded in gray. (F) The proportion consistent with the temperature-sensing hypothesis increased as the slope factor increased. The slope factor was calculated by fitting SCH and ΔH values to a linear function for each site with buried/exposed changes. The hydrophobicity scale of SCH was determined by Hessa et al (represented in blue) and Moon et al (represented in green), respectively. The x-axis represents the proportion that aligns with the temperature sensitivity hypothesis. The sites located in the first and third quadrants from panel (E) were mapped onto the cryo-EM structure of TRPM8 (PDB ID: 7WRA) with their sidechains shown in blue. (G) Representative whole-cell current recordings of TRPM8 mutants that regulate the cold-activated properties of TRPM8 in response to cold. Cold- and menthol-induced currents were measured at +80 mV to record TRPM8 channel activation. To estimate the open probability (Po), current amplitudes under saturating menthol stimulation (1 mM) were normalized, assuming maximal channel opening under these conditions. Source data are available online for this figure.

To validate the HRF-MS results, we incorporated the fluorescent unnatural amino acid ANAP into TRPM8. A shift of the ANAP emission peak to a longer wavelength indicates that the residue becomes more exposed to a hydrophilic environment (Chatterjee et al, 2013). Although many sites in TRPM8 did not tolerate ANAP incorporation (Xu et al, 2020; Yang et al, 2020), some ANAP-incorporated mutants remained functional and showed shifts in emission spectra during cold activation. Residues W436 and M1059 were identified in both HRF-MS and ANAP experiments. Their FC and shifts in the ANAP emission peak were consistent with each other (Fig. EV2C–E). Therefore, our ANAP imaging data for TRPM8 corroborates the validity of the HRF-MS results.

## Functional characterization in TRPM8 mutants

With the identification of residues in TRPM8 that displayed changes in cold activation related to their buried or exposed states, we investigated whether altering SCH at these sites could specifically influence cold sensitivity. We mutated each of the residues identified through HRF-MS (Fig. 1C; Table EV1) to be either more hydrophobic or more hydrophilic; only those that exhibited functional currents were included in electrophysiological analysis (Dataset EV3).

We then measured the cold activation of all the mutants by patch-clamp recording (Figs. 2 and EV3). For instance, from HRF-MS experiments, W137 exhibited increased exposure during cold activation, with a FC value of 1.25 (Fig. 1C). When the W137 residue was mutated to increase SCH (W137L), the absolute ΔH values of the mutants became more negative, indicating increased cold sensitivity. In contrast, the W137H mutants with decreased SCH exhibited reduced absolute ΔH values and reduced cold sensitivity (Fig. 2A). The relationship between hydrophobicity and ΔH was quantified as the slope of a linear fit (Fig. 2B); a positive slope factor supports our hypothesis on temperature sensing. In contrast, residue W651 became more buried during cold activation, with an FC value of 0.87 (Fig. 1C), and its negative slope (Fig. 2C,D) is consistent with our hypothesis. However, it is important to note that not all residues exhibited a strong linear correlation between hydrophobicity and ΔH. This is likely because changes in

residue exposure represent one important, but not the only, contributing factor influencing temperature sensitivity. Therefore, the observed slope factor should be understood as a statistical measure reflecting the trend by which residue hydrophobicity influences temperature sensitivity.

The hydrophobicity scale of SCH was initially determined by Hessa et al (Hessa et al, 2005) and Moon et al (Moon and Fleming, 2011). To fully establish the relationship between SCH and the water–protein interaction-based temperature sensing hypothesis, we analyzed each site identified in HRF-MS using two different types of hydrophobicity scales (Figs. 2 and EV3). We found that not all residues exhibited consistent trends across the two hydrophobicity scoring systems. This discrepancy can be explained by two main factors. First, certain amino acids are inherently insensitive to temperature stimuli, leading to low slope factors; in addition, differences in hydrophobicity values assigned by different SCH scales introduce variability into slope factor measurements. Second, mutations designed to alter the apparent hydrophobicity may render the channels unresponsive to cold or menthol, thereby preventing functional data collection and compromising the accuracy of slope factor measurements. Residues with similar hydrophobicity-ΔH trends were selected for further analysis. Subsequent plotting of FC against slope factors revealed that the majority of residues undergoing buried/exposed changes conformed to the water–protein interaction-based temperature-sensing hypothesis (Fig. 2E,F).

To understand the apparent discrepancy, we performed more in-depth analyses. Our temperature-sensing hypothesis suggests that steeper slopes indicate a stronger contribution of a residue to temperature sensitivity. When residues with larger slopes were examined, the proportion consistent with the water–protein interaction model increased markedly, from 59.2 (slope factor >0) to 88.8% (slope factor >10) (Hessa et al, 2005) or from 59.2 to 77.8% (Moon and Fleming, 2011) with the two different hydrophobicity scales of amino acids (Figs. 2E,F and EV4A,B). By either hydrophobicity scale, residues with the largest slope factors closely followed the predictions of the water–protein interaction hypothesis and exerted a decisive influence on cold sensitivity. (Fig. 2G; Table EV2). In contrast, residues that did not align with the hypothesis failed to modulate cold sensitivity (Fig. EV4C).

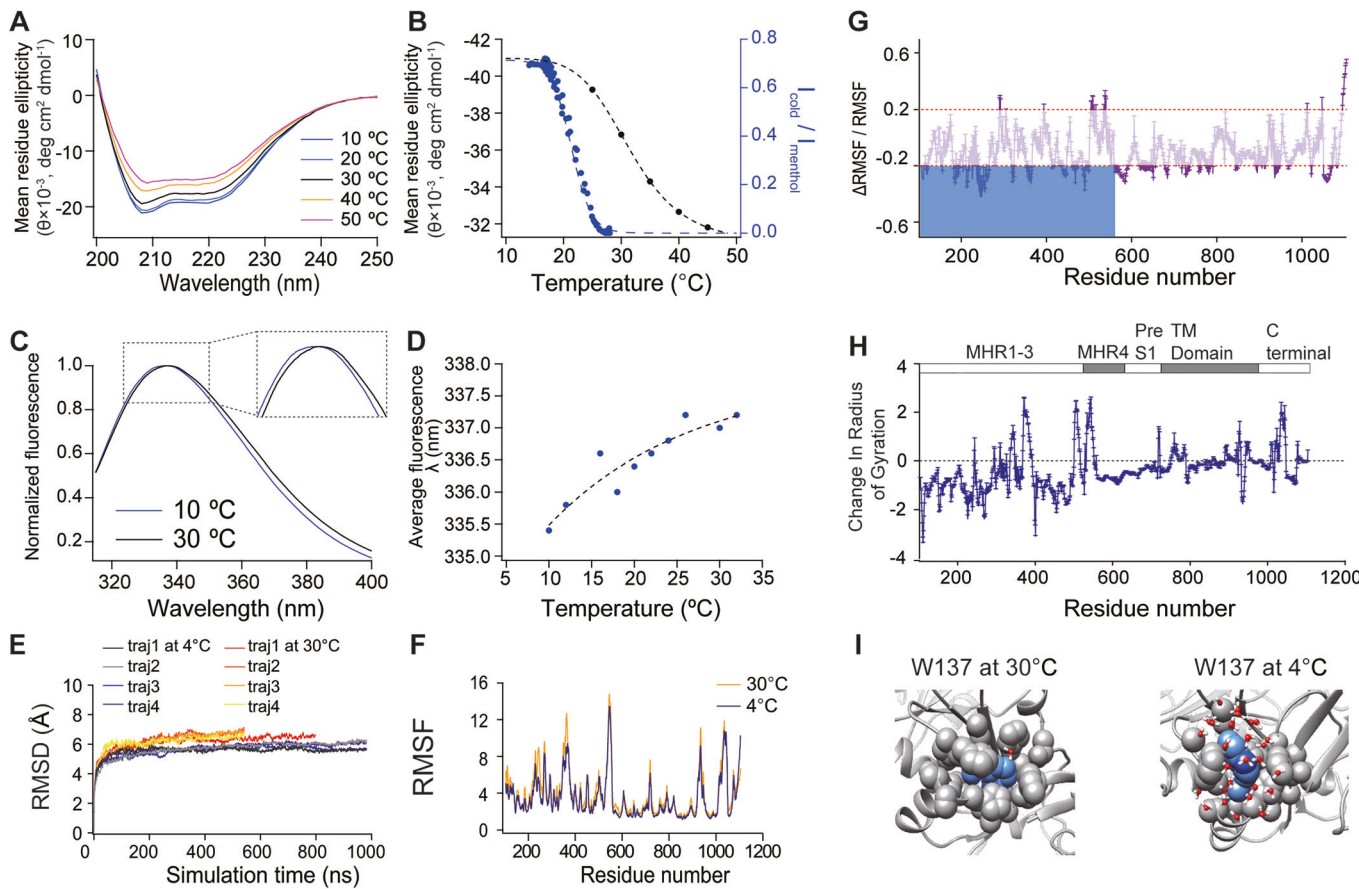

**Figure 3.  Water–protein interactions for temperature sensing.**

(A) Representative CD spectra of the MHR1-3 domains in TRPM8 measured at different temperature levels. (B) Comparison of cold activation of TRPM8 current normalized to menthol activation (dots in blue) and changes in mean residue ellipticity (dots in black). (C) Representative intrinsic tryptophan emission spectra of the MHR1-3 domains measured at different temperature levels. The inlet showed the cooling-induced shifts in emission peak. (D) Temperature dependence of intrinsic tryptophan emission peaks values. (E) RMSD (relative to the TRPM8 structure in the apo state) plots of MD simulation trajectories at either 4 °C or 30 °C. All three RMSD curves stabilize within 100 ns. (F) The RMSF changes of TRPM8 were observed at 4 and 30 °C, respectively. (G) The fractional changes in RMSF (RMSF at 4 minus 30 °C) normalized to RMSF at 4 °C. Dashed lines indicated the 20% significance in fluctuation level, below which any fluctuation was due to nonspecific thermal effects (Wen and Zheng, 2018). (H) The temperature-dependent changes (4 °C minus 30 °C) in the radius of gyration (Rg) for each residue in TRPM8. (I) Representative snapshots of MD trajectories showing the buried/exposed state of W137 (residue in blue) and the water molecules nearby (oxygen and hydrogen atoms in water colored in red and white, respectively) at 4 and 30 °C. Source data are available online for this figure.

When we mapped all these cold-sensitive sites onto the TRPM8 channel structure, we observed that they were clustered within the N-terminal MHR1-3 domains (Figs. 2F and EV4B), while other residues were located in different regions of TRPM8. Such a distinct pattern in the spatial distribution of residues further prompted us to investigate the MHR1-3 domains.

## MHR1-3 domains directly responded to cold

The clustered distribution of highly temperature-sensitive residues located in MHR1-3 domains, suggesting the cluster is critical for cold sensing in TRPM8 (Fig. 2F). If the MHR1-3 domains directly sense cold, these domains should show cold-induced conformational changes. To test this hypothesis, we first expressed and purified the MHR1-3 domains (Fig. EV5A,B, residue 109 to 500 in blue). We then monitored the cooling-induced conformational changes in MHR1-3 domains by either far-UV circular dichroism

(CD) (Fig. 3A,B) or intrinsic tryptophan fluorescence measurements (Fig. 3C,D), which are classic strategies widely used to study thermodynamic properties of domains in TRP channels (Kim et al, 2020) and other proteins (Greenfield, 2006). The CD spectra of MHR1-3 domains exhibited two minima at 208 and 222 nm, which is characteristic for α-helix containing proteins like MHR1-3 domains (Fig. 3A). Cooling to 10 °C induced changes in CD spectra, which displayed two-state behavior (Fig. 3A). Fitting the data to a two-state sigmoidal model as in a previous study (Kim et al, 2020) yielded a ΔH of $-44 \pm 2$ kcal/mol and an apparent unfolding midpoint temperature of $31.3 \pm 0.3$ °C (Fig. 3B, dots and curve in black). ΔH values measured from TRPM8 open probability in previous studies ranged from $-57.8$ kcal/mol to $-112$ kcal/mol (Brauchi et al, 2004; Yang et al, 2010), so it indicates that assuming a full cooperativity, three or fewer copies of MHR1-3 domains in the tetrameric TRPM8 channel are sufficient to account for the enthalpic changes in cold activation. Moreover, the apparent

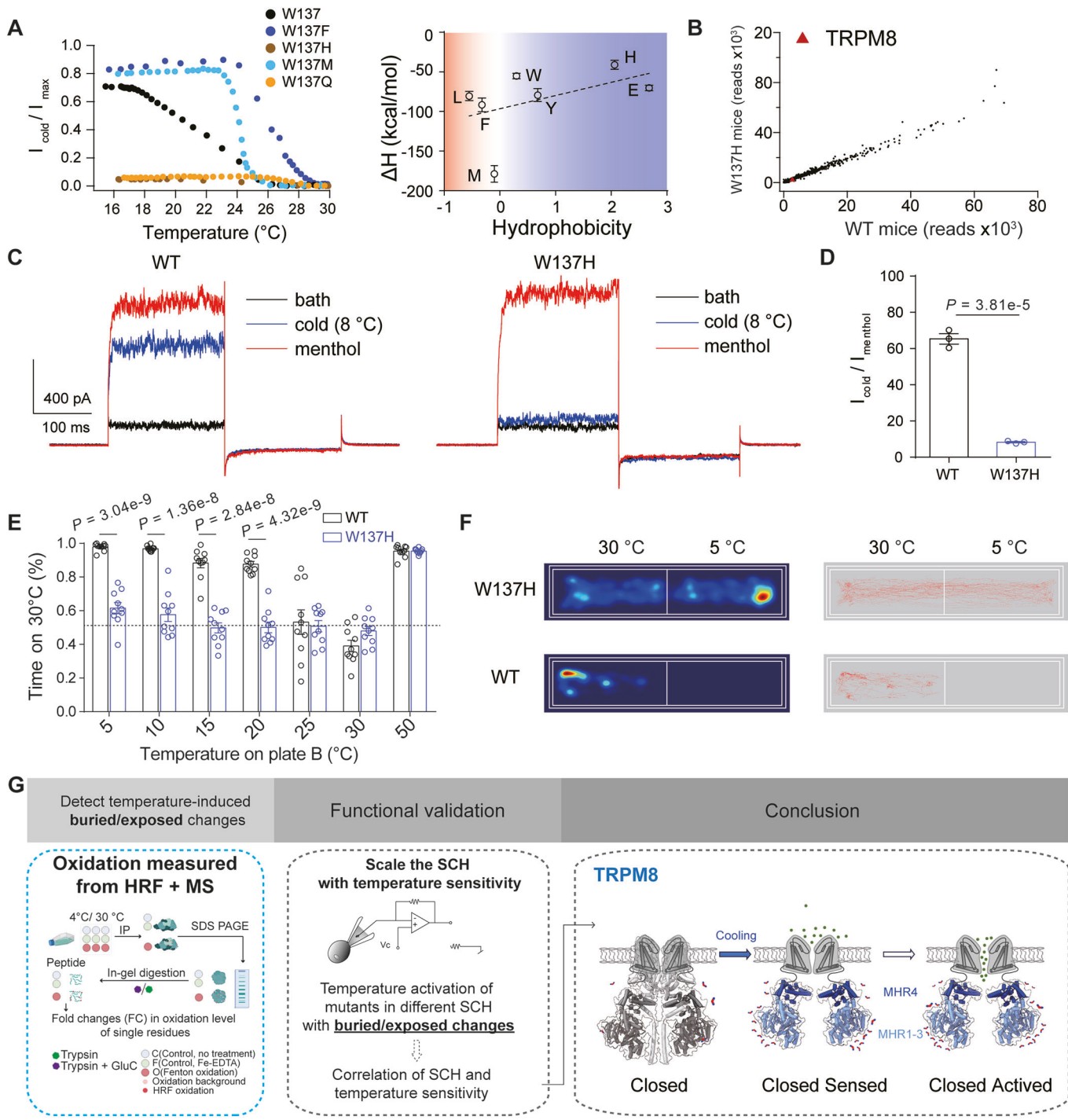

unfolding midpoint temperature of MHR1-3 domains was between the normal body temperature of mammals like mouse (~36.5 °C) and the cold activation threshold of TRPM8 (~28 °C) (Fig. 3B, dots and curve in blue), thus enabling the prompt detection of temperature drops.

We further measured the temperature dependence of intrinsic tryptophan emission. There are eight tryptophan residues in MHR1-3 domains, so if temperature induces conformational changes in MHR1-3 domains to alter the local chemical environment of these residues, their emission peak will shift (Royer, 2006). We observed cooling from 30 to 10 °C induced blue shifts in intrinsic tryptophan emission (Fig. 3C,D). In contrast, when the N-terminal ankyrin-repeat-like domain (ARD) of the heat-sensitive TRPV2 channel was expressed and purified (Fig. EV5C,D), within the same 30 to 10 °C temperature range, we did not observe temperature-dependent changes in either far-UV CD spectra or intrinsic tryptophan emission (Fig. EV5E–H). Therefore, our observations suggested that the MHR1-3 domains protein,

**Figure 4.  Tuning of SCH by the W137H mutation largely reduced cold sensitivity in mice.**

(A) Representative whole-cell current recordings of cold activation for W137 and its mutants, along with the correlation between SCH and ΔH values for the W137 residue. The hydrophobicity scale of SCH was determined by Hessa et al. (B) Transcriptome-wide comparison of mRNA expression between wild-type and W137H mice. (C) Representative inside-out patch recordings of DRG neurons in wild-type (left panel) and W137H (right panel) mutant mice activated by cold and 1 mM menthol. ($n = 3$ biological replicates; statistical significance was determined using a two-sided Student's $t$-test. Data were presented as mean ± s.e.m.). (D) Normalized currents of DRG neurons in wild-type and W137H mutant mice. ($n = 3$ biological replicates; statistical significance was determined using a two-sided Student's $t$-test. Data were presented as mean ± s.e.m.). (E) Mice were allowed to move freely in a two-temperature choice test with a control plate (30 °C) and a test plate (ranging from 5 to 30 °C). The percentage of time spent at the control plate was measured every 3 min ($n = 10$ biological replicates; statistical significance was determined using a two-sided Student's $t$-test. Data were presented as mean ± s.e.m.). (F) Representative heat maps (left panel) and traces (right panel) of WT and W137H mice in two-temperature choice assays. (G) Workflow of the hypothesis-driven study on temperature-induced mechanisms in the TRPM8 channel. The workflow of this hypothesis-driven study, where the hypothesis testing was composed of two major steps: the detection of temperature-induced buried/exposed changes and the functional validation. Upon cold activation of TRPM8, the intracellular MHR regions were the first to sense the temperature change, initiating conformational shifts in the channel that result in its opening. Source data are available online for this figure.

which hosted the residues changing their buried/exposed state to modulate cold sensitivity in TRPM8 (Fig. 2F), directly responded to cold with a sufficiently large enthalpic change for cold activation. This is also supported by the study of TRPM8 evolution, which shows that the formation of the functional MHR1-3 domains in TRPM8 bestowed the channel with cold sensitivity (Lu et al, 2022).

## Energetics in water–protein interactions for temperature activation

To further quantify whether the energetic changes associated with water–protein interactions in TRPM8 are sufficient for temperature activation, we investigated water–protein interactions at residues that effectively regulate temperature sensitivity in the MHR1-3 domains. Due to the limited resolution of the cryo-EM structure of the TRPM8 channel (Diver et al, 2019; Yin et al, 2019; Yin et al, 2018; Zhao et al, 2022), water molecules cannot be directly observed in the recently published structures. Instead, we performed multiple all-atom molecular dynamics (MD) simulations using the apo state structure (PDB ID: 7WRA) at either 4 or 30 °C with a total simulation length of over 6 μs (Fig. 3E). Though the TRPM8 channel was not directly opened by low temperature during the simulations, we gained unique insights into the cold-sensing mechanisms.

MD simulation corroborated our previous findings in the critical role of MHR1-3 domains in cold activation of TRPM8. Interestingly, we observed that in most domains, the root mean square fluctuation (RMSF) at 4 °C was smaller than those at 30 °C (Fig. 3F; Dataset EV4), indicating that lowering the temperature reduced the mobility of atoms in TRPM8. The MHR1-3 domains showed the most significant decrease (Fig. 3G, shaded area in blue), which was larger than a stringent 20% threshold of nonspecific thermal effect (Wen and Zheng, 2018) (Fig. 3G, shaded area in gray between solid lines in red). We also measured the changes in radius of gyration ($R_g$) between 4 and 30 °C (Fig. 3H). The negative changes in $R_g$, which suggested the residues became more packed, were also clustered in the MHR1-3 domains (Fig. 3H).

More importantly, we calculated the changes in protein–water electrostatic interaction energy and van der Waals (vdw) energy of the TRPM8 simulation system. We observed a large decrease in the protein–water electrostatic interaction energy (about −84.8 kcal/mol after being scaled by a factor of 80, which is the typical value of dielectric constant in proteins (Wen and Zheng, 2018) at 4 °C as compared to that at 30 °C (Table EV3). Interestingly, when the

protein–water electrostatic interaction energy in the last 100 ns of simulation was compared to that of the first 100 ns simulation at 4 °C, only in MHR1-3 domains we observed a time dependent decrease in protein–water electrostatic energy (−36.9 kcal/mol), while MHR4 and the rest of TRPM8 protein showed an increase (13.8 and 15.6 kcal/mol, respectively). In comparison, at 30 °C, the change in protein–water electrostatic energy of MHR1-3 domains was much smaller (−12.3 kcal/mol), indicating such changes were induced by the low temperature (Dataset EV5). Therefore, our MD results again suggested that the MHR1-3 domains, but not MHR4 or other domains in TRPM8, are directly responding to cold.

Furthermore, we counted the changes in the number of water molecules contacting TRPM8 residues in MD (Table EV4) and compared the values to FC measured in HRF-MS. For the residues whose cold-sensitivity was measured, the changes in the number of contacting water molecules predicted by MD are also consistent with the temperature-sensing hypothesis. For instance, MD results suggested that W137 was predominantly buried at 30 °C. However, cooling to 4 °C induced conformational rearrangements, exposing W137 to a more hydrophilic environment, with approximately eight additional water molecules in contact (Fig. 3I; Table EV4). The FC of W137 measured in HRF-MS was 1.25 (Table EV1; Datasets EV1 and 2), again indicating a buried to exposed change by cooling at this site. Assuming hydrogen bond strength ranges from 1.5 to 9.5 kcal/mol (Franks, 2000; Sheu et al, 2003; Yang et al, 2015) and there is only one hydrogen bond formation/breaking per water molecule, a change of ten water molecules may cause an energetic change of 15 to 95 kcal/mol. Therefore, in the tetrameric TRPM8 channel with four copies of buried/exposed residues, the estimated energy changes associated with changes in water count (48 to 304 kcal/mol) is sufficient for the ΔH as measured from TRPM8 open probability in previous studies (Brauchi et al, 2004; Yang et al, 2010).

## W137H-*trpm8* mice exhibited much reduced cold sensitivity

To further validate our hypothesis, we mutated W137 (Fig. 4A), to all the other 19 residues and characterized each of the mutants by patch-clamp recording. We found that only eight mutants at site 137 remained to be functional (Table EV5). Like our previous observation (Fig. 2A,B), most of the mutants followed the prediction from our hypothesis: decreasing SCH at these exposed sites caused reduction in cold sensitivity (Fig. 4A). For instance,

W137H mutant was activated by menthol (Table EV5), but its cold sensitivity, as reflected in ΔH of cold activation, was much reduced (Fig. 4A). Therefore, based on the comprehensive observations outlined above, we assert that the water–protein interactions are a critical and integral component of the cold-sensing mechanism in TRPM8.

We then investigated whether altering the side-chain hydrophobicity of residues in TRP channels can specifically modulate the temperature sensitivity of the channel in vivo. To do this, we introduced a single point mutation W137H, which exhibited normal menthol activation but little cold activation (Fig. 4A; Table EV5), to Trpm8 in transgenic mice (W137H-trpm8 mice) by CRISPR-Cas9 technology. W137H homozygous mutant mice showed similar transcription levels of Trpm8 compared with wild-type (WT) mice (Fig. 4B). As expected, the TRPM8 channel was well-expressed in dorsal root ganglion isolated from either WT mice or W137H-trpm8 mice and could still be activated by 1 mM menthol in patch-clamp recordings. In contrast, dorsal root ganglion (DRG) neurons from the W137H-trpm8 mice could barely be activated by cold (Fig. 4C,D).

We then employed a two-temperature choice assay to determine whether W137H-trpm8 mice have deficits in temperature discrimination. As expected, WT mice did not prefer 30 °C control plate over 25 °C test plate, but they moved almost exclusively on the 30 °C control plate over the 5 °C test plate (Fig. 4E,F). Compared with WT mice, W137H-trpm8 mice showed severe deficits in detecting cool temperatures, as they stayed on either the control or test plate with nearly equal proportions of time. Therefore, these results demonstrated that rationally changing SCH of a single amino acid with the buried/exposed state changes (therefore the water–protein interactions) in TRPM8 (W137H) could alter the cold-activation properties of this channel, which further virtually abolished cold-sensing behaviors in mice.

In this study, we tested the hypothesis of water–protein interactions underlying the temperature sensing in TRPM8 by comprehensively searching for residues with burial/exposure changes in temperature activation (Fig. 4G). We systematically altered the SCH in all the identified residues, and then measured the impact of SCH on temperature sensitivity in mutants by patch-clamp recording (Figs. 2 and EV3). As for TRPM8, these experiments led us to identify the MHR1-3 domains that directly responded to cold, which exhibited clear changes in interactions with water molecules and sufficiently large energetic changes for cold activation. Therefore, we believe that the MHR1-3 domains, especially the residues with burial/exposure changes in these domains, serve as the driving "engine" for cold sensing in TRPM8.

Nevertheless, the residues supporting our hypothesis but distributed outside the intracellular MHR1-3 domains (Figs. 1C and 2F) indicated that a more complex nature of cold sensing and activation mechanism. Indeed, by analyzing the TRPM8 channel in various vertebrates, previous studies showed that residues in the transmembrane domains can largely modulate cold activation properties (Matos-Cruz et al, 2017; Pertusa et al, 2018; Yang et al, 2020). However, these analyses were limited to evolutionarily advanced vertebrate species, where cold activation of TRPM8 was modulated but not abolished. As MHR1-3 domains serve as the engine for cold sensing in TRPM8, these domains may also be critical for molecular evolution. Given the central role of the MHR1-3 domains as the cold-sensing engine and their evolutionary

emergence conferring cold sensitivity during terrestrial tetrapod evolution (Lu et al, 2022), we propose that residues outside the MHR1-3 domains primarily serve as modulators of cold sensing and activation in TRPM8.

Methodologically, our HRF-MS experiments detected approximately 84% of amino acid residues, reflecting inherent limitations of hydroxyl radical labeling using the Fenton chemistry system (Liu et al, 2020a). Recent developments, including pulsed discharge lamps (Sharp et al, 2021) and plasma-based systems (Minkoff et al, 2017), offer significantly higher temporal resolution for hydroxyl radical production. The adoption of these faster labeling techniques is expected to reduce labeling-induced conformational artifacts and improve the accuracy and reliability of footprinting data in future studies.

By systematically screening residues with burial/exposure changes during cold activation, we identified specific amino acids that regulate temperature sensitivity, which are consistent with the water–protein interaction hypothesis. We propose that not all amino acids with dynamic buried/exposed conformational changes are involved in temperature sensing; rather, the buried/exposed states of specific amino acids (including water–protein interactions) are key components of the temperature-sensing mechanism, providing valuable insights into the temperature-sensing mechanisms of TRP channels.

Understanding the temperature-sensing mechanisms in TRPM8 is crucial for translational applications, as this channel plays a key role in pain detection and is a target for analgesic drugs (Julius, 2013). However, many blockers of these channels developed for analgesia caused changes in body temperature and blunting of acute temperature sensation in patients, leading to failures in clinical trials (Gavva, 2009; Kort and Kym, 2012). By elucidating the temperature-sensing mechanisms and temperature-induced conformational changes, we can aid in the development of modality-specific blockers, such as the cyclic peptide DeC-1.2, which we recently designed to inhibit ligand gating of TRPM8 without affecting cold activation (Aierken et al, 2021), thus minimizing adverse side effects while retaining analgesic efficacy.

## Methods

**Reagents and tools table**

| Reagent/resource | Reference or source | Identifier or catalog number |
|---|---|---|
| **Experimental models** | | |
| HEK-293 cells (H.sapiens) | ATCC | CRL-1573 |
| C57BL/6J (M. musculus) | Cyagen Biosciences, Inc. | N/A |
| **Recombinant DNA** | | |
| pEGFP-N1-TRPM8 | This study | N/A |
| pEG BacMam-TRPM8-strep | This study | N/A |
| **Antibodies** | | |
| Mouse Anti Strep-Tag II Antibody, mAb | Detaibio, Inc. | DTA0211 |
| **Oligonucleotides and sequence-based reagents** | | |
| PCR primers | This study | Dataset EV3 |

| Reagent/resource | Reference or source | Identifier or catalog number |
|---|---|---|
| **Chemicals, enzymes, and other reagents** | | |
| SDS-PAGE | GeneScript | M00653, M00654 |
| Fetal Bovine Serum (FBS) | Noverse | NFBS-2500A |
| 0.25% Trypsin | Cienry | CR-27250 |
| Trypsin | Sigma | T1426 |
| collagenase | Sigma | C2674 |
| DNase | Sigma | DN25 |
| Trypsin Inhibitor from soybean | Sigma | 10109886001 |
| DMEM basic | gibco | 11971025 |
| (-)-Menthol | TCI AMERICA | M0545 |
| Penicillin-Streptomycin Solution, 100 × | Cienry | CR-15140 |
| Tris-HCl (100 mM, pH 8.0) | Sangon Biotech | B548127 |
| Triton X-100 | Sangon Biotech | A110694-0100 |
| Na$_2$ EDTA | MERCK | E5134 |
| 1% deoxycholate | MERCK | 264103 |
| Streptactin Bead 4FF antibody | Smart-Lifesciences | SA053005 |
| Coomassie blue | Sangon Biotech | E607056-0250 |
| Trypsin digestion | Hualishi Technolog | HLS TRY001C |
| Glu-C | Hualishi Technology | HLS GLU001 |
| C18 spin tips | Thermo Scientific | Pierce™ 84850 |
| 2×Taq Plus Master Mix | Vazyme | P213-01 |
| Lipofectamine 3000 | Life technologies | L3000015 |
| **Software** | | |
| Graphpad Prism | GraphPad Software Inc. | Version 9.5.0 |
| Adobe Illustrator | Adobe | Version 2022 |
| Igor Pro | Igor Pro | Version 6.11 |

## Cell treatment and Fenton oxidation in vivo

The transfected HEK cells were stimulated in three different conditions for 5 min: (i), C for Ctrl, HEK cells were bathed in 5 ml of 1x PBS, no other treatment; (ii), F for Fe treatment, namely HEK cells were bathed with 10 mM Fe(II)-bound EDTA in 5 ml 1x PBS; and (iii), O for OH radical oxidation of cells, which means that HEK cells were oxidized by Fenton oxidation in vivo. The cell density of these above conditions was around $10^7$ cells/ml as optimized previously (23). The oxidation process was slightly modified from OMPF footprinting as published previously (Zhu et al, 2009b). Firstly, HEK cells were resuspended and bathed in 5 ml 1x PBS. Then 10 mM (NH4)$_2$Fe(SO$_4$)$_2$ and 25 mM Na$_2$EDTA were added into the system and mixed by brief vortexing, then 70 μL of 30% H$_2$O$_2$ were added to make a final concentration of 0.3% H$_2$O$_2$ into the suspension for Fenton oxidation, lasting for 5 min. Two temperatures were included in this study for oxidation under ion channel opening (4 °C) and ion channel closure (30 °C). The reaction was quenched after adding an equal volume of Tris-

HCl (100 mM, pH 8.0, Sangon Biotech). The cells were collected by centrifugation at 1100 rpm, room temperature, for 10 min for later processes. For each condition, three biological replicates were conducted. As a summary, HEK cells treated under three biological conditions at two temperatures were included in this study.

## Immunoprecipitation (IP), SDS-PAGE separation, and in-gel digestion

The detailed protocols for IP, SDS-PAGE separation, and in-gel digestion were specified as previously published (Zhu et al, 2011; Zhu et al, 2017). Briefly, HEK293 cells were transfected with TRPM8 alone or co-transfected with TRPM8 and the phosphoinositide-interacting regulator of TRP (Pirt) using Lipofectamine 2000 (Thermo Scientific). Pirt is a protein with two helical transmembrane domains, which is specifically expressed in the peripheral nervous system. It is known to bind and modulate TRPV1 (Kim et al, 2008) and TRPM8 channel (Tang et al, 2013) activity and expression. Its co-expression improves the accuracy of downstream analyses. After treatment, HEK cells were lysed in the in-house prepared lysis buffer (1% Triton X-100, 1% deoxycholate, 0.1% SDS, 150 mM NaCl in 50 mM Tris-HCl, pH 7.4). Thereafter, Streptactin Bead 4FF antibody (Smart-Lifesciences) was added to the supernatant overnight at 4 °C. The bound complexes were pulled down and were subsequently subjected to electrophoresis on an 8% SDS-PAGE. After SDS-PAGE separation, the page was stained with Coomassie blue (Sangon Biotech) to locate the position of the TRPM8 protein.

For the identification of TRPM8 and its binding partners, the whole lane was cut into ten fractions for later in-gel digestion steps. For identification and quantification of TRPM8 only in the latter experiment, only the area around 130 kDa from the lane was selected and cut into 2–3 gel bands. They were subsequently digested into peptides either by (i) a two-time trypsin digestion (Hualishi Technology, enzyme: protein ratio at 1:100 each time, 4 h plus 12 h intervals) at 37 °C overnight; or (ii) a sequential two-enzyme digestion first with trypsin (enzyme: protein ratio, 1:100) at 37 °C for 4 h, and second with Glu-C (Hualishi Technology, (enzyme: protein ratio, 1:100)) at 37 °C overnight. The peptides were extracted, and desalted using C18 spin tips (Thermo Scientific) according to the manufacturer's protocol.

## LC-MS/MS analysis

The LC − MS/MS analysis was performed either on a QE-HF or QE-HF-X mass spectrometer (Thermo Fischer) with the same MS setting (Dataset EV1) coupled with an UltiMate 3000 liquid chromatography system (Thermo Fisher). Around 500 ng of peptides prepared from each fraction were separated in a fused silica column (15 cm × 75 μm ID, National Institute of Biological Sciences) packed with C18 (1.9 μm 100 Å) at a flow rate of 300 nL/min. Buffer A (2% ACN, 0.1% FA) and buffer B (80% ACN, 0.1% FA) were used for the LC gradient. Nano spray (NSI) was applied for ionization.

For library generation, firstly, DDA MS data acquisition for the ten fractions of TRPM8 complex was performed at an effective 60 min LC gradient that ramped from 3 to 28% buffer B. This is to identify TRPM8 complex proteins. Secondly, DDA MS data acquisition for the 2–3 fractions around 130kD area of both oxidized and non-oxidized TRPM8 proteins was performed at a longer LC gradient, namely, an effective 90 min LC gradient that ramped from 3 to 28% buffer B in

90 min (Zhu et al, 2020b). This is to better characterize TRPM8 oxidation sites. A shotgun DDA MS strategy was applied with the following parameters: The survey scan (400–1200 m/z) was acquired at a resolution of 60k, a maximum ion accumulation time of 80 ms, and an AGC target of 3e6. The top 20 ions were selected for MS2 analysis at a resolution of 30k, a maximum ion accumulation time of 100 ms, and an AGC target of 1e5. Dynamic exclusion was activated with a duration of 30 s. Ions with a charge state of +1 and >5 were excluded from MS/MS. The isolation width was set to 1.6 Da, and the normalized collision energy (NCE) was set at 27. Spectra were acquired in centroid format.

For proteomic identification and quantification of oxidized domains, TRPM8 peptides were spiked with 10% iRT peptides (Biognosys) for later calibration. DIA MS data acquisition was performed at an effective 60 min LC gradient that ramped from 3 to 28% buffer B. A DIA MS strategy was applied (Zhu et al, 2020b). The survey scan (390–1010 m/z) was acquired at a resolution of 60k, a maximum ion accumulation time of 80 ms, and an AGC target of 3e6. The MS2 scan was acquired in 24 windows set as 389.5–410.5, 409.5–430.5, 429.5–450.5, …-749.5–770.5, 769.5–830.5, 829.5–890.5, 889.5–950.5, 949.5–1010.5. The resolution for MS2 was set as 3E4, with an automated maximum ion accumulation time, and an AGC target of 3E4. The normalized collision energy (NCE) was set at 28. Spectra were acquired in centroid format.

## TRPM8 spectral library generation

In total, 45 DDA files were acquired for library generation, including 28 using trypsin and 17 using trypsin coupled with Glu-C as digestion enzymes, respectively. We also included 28 test samples of TRPM8 under different oxidation conditions during the experiment to increase sequence coverage, as labeled in Dataset EV1.

The ten DDA files of the TRPM8 complex were searched against the fasta sequence composed of the human reviewed fasta, the Pirt sequence, and the *Mus musculus* TRPM8 sequence, using pFind software with the default settings and an FDR cutoff of 0.01. Proteins that do not interact with TRPM8 (KEGG: mmu04750) and fulfill one of the following criteria were excluded from the resultant matrix: (1) with only one identified peptide; (2) with multiple matches (protein groups); (3) keratin. The filtered result was transferred to a filtered FASTA database.

The 35 DDA files of TRPM8 under different treatment conditions were searched against the filtered fasta database, using pFind software with Open Search settings and an FDR cutoff of 0.01. The following oxidation modifications were considered as variable modifications with maximal modification number per peptide as three:16Ox[M];16Ox[D];16Ox[F];16Ox[H];16Ox[-K];16Ox[N];16Ox[P];16Ox[R];16Ox[W];16Ox[Y];32Ox[C];32Ox[-F];32Ox[M];32Ox[W];32Ox[Y];14Ox[I];14Ox[E];14Ox[K];14Ox[-L];14Ox[P]. Other parameters were set as default. Proteins with modifications other than oxidation, carbamidomethyl, or deamination were excluded from the resultant matrix. The filtered result contains 1777 peptide precursors, including 810 oxidized sites and was used as the TRPM8 library.

## DIA MS quantification of oxidized peptides of TRPM8

A total of 72 DIA files were used for the DIA quantification, including 36 using trypsin and 36 files using trypsin coupled with Glu-C as digestion enzymes, respectively. The DIA files of TRPM8 were analyzed using Skyline (version: daily-20.2.1.215). The TRPM8 library was imported for spectra match. We performed Open Search with pFind, which would allow all possible modifications to be considered. iRT were imported to calibrate retention time and MS error PPM was set to be lower than 15 ppm. Other parameters were set as default. All fragments were selected for quantification of its origin peptide precursor. For each peptide, the raw intensity at both MS1 and MS2 were added together. A resultant peptide matrix was generated as shown in Dataset EV1.

The oxidation level was calculated by:

$$Oxi\% = Intensity_{Oxi}/(Intensity_{Oxi} + Intensity_{nake})$$

The relative oxidation level was calculated by:

$$Relative\ Oxi\% = Oxi\%_{oxi}/(Oxi\%_{ctrl1} + Oxi\%_{ctrl2})$$

The fold changes were calculated by:

$$Fold\ change = Relative\ Oxi\%_{4\,°C}/Relative\ Oxi\%_{30\,°C}$$

The selected range of FC between 0.83 and 1.2 (representing a ±20% change) is based on established conventions in previous HRF-MS studies (Zhu et al, 2009a; Zhu et al, 2011). This threshold takes into account the typical background noise and inter-replicate variability inherent in mass spectrometry measurements, and serves as a practical cutoff to distinguish biologically meaningful conformational changes.

## Gene synthesis and mutagenesis

*Mus musculus trpm8* (GeneID: 171382) were synthesized by Tsingke (Beijing, China) based on the predicted gene sequence and subcloned into the pEGFP-N1 vector. Point mutations were constructed using 2×Taq Plus Master Mix (Vazyme) following the manufacturer's instructions. All point mutations were summarized in Dataset EV3 and confirmed by sequencing.

## Cell transient transfection

HEK293T cells were cultured in Dulbecco's modified Eagle medium (DMEM) supplemented with 20 mM L-glutamine and 10% fetal bovine serum, incubated at 37 °C with 5% CO₂. Cells were transiently transfected by Lipofectamine 3000 (Life Technologies) following the manufacturer's protocol. Patch-clamp recordings were performed 24 h after transfection.

## Isolation of DRG Neurons

Neuronal somata were isolated using enzymatic and mechanical dissociation. Adult male rats were sacrificed by decapitation, and DRGs were dissected from the upper lumbar to mid-thoracic regions of the vertebral column. The DRGs were minced into small pieces using iridectomy scissors and transferred to a flask containing 5 ml DMEM supplemented with trypsin (10,000 U/ml, Sigma), collagenase (1 mg/ml, Sigma), and DNase (0.1 mg/ml, Sigma). The tissue was incubated at 35 °C for 50–60 min, after which soybean trypsin inhibitor (Sigma) was added in sufficient quantity to neutralize the trypsin activity.

Following enzymatic digestion, the ganglia were gently triturated to disperse individual neurons, which were then plated in uncoated Petri dishes. The neurons were used for

electrophysiological recordings within 8 h after isolation, as required by the experimental design.

## Electrophysiology

Patch-clamp recordings were performed with a HEKA EPC10 amplifier controlled by PatchMaster software (HEKA). Whole-cell recordings were performed using a voltage-step protocol at ±80 mV. Patch pipettes were prepared from borosilicate glass and fire-polished to a resistance of ~4 MΩ. For whole-cell recording, serial resistance was compensated by 60%. A solution with 130 mM NaCl, 10 mM glucose, 0.2 mM EDTA and 3 mM Hepes (pH 7.2) was used in both bath and pipette for whole-cell recordings. Current was sampled at 10 kHz and filtered at 2.9 kHz. Cold- and menthol-induced currents were measured at +80 mV to record TRPM8 channel activation. To estimate the open probability (Po), current amplitudes under saturating menthol stimulation (1 mM) were normalized, assuming maximal channel opening under these conditions.

Gravity-driven system (RSC-200, Bio-Logic) was used to perfuse bath solution or menthol onto the cell membrane. Bath and ligand solution were delivered through separate tubes to minimize the mixing of solutions. The patch pipette was placed in front of the perfusion tube outlet.

## Temperature control

We used a precooled bath solution to activate TRPM8. Cells were recorded and placed in a bath solution at over 30 °C before recording. A TA-29 miniature bead thermistor (Harvard Apparatus) was placed right next to the pipette to ensure accurate monitoring of local temperature.

## Calculation of ΔH and ΔS

To calculate the change of enthalpic (ΔH) and the change of entropic (ΔS) due to the temperature-driven transition, we constructed Van't Hoff plots and fitted them with the equation:

$$\ln Keq = \frac{-\Delta H}{RT} + \frac{\Delta S}{R}$$

where R represents the gas constant, T represents the temperature in Kelvin, Keq represents the equilibrium constant.

## Protein expression and purification

Dialysis-based cell-free protein synthesis reactions were performed using modified *E.coli* cell extracts, amino acid mix and reaction mix, etc., as described in the manufacturer's (GZL Bioscience Co. Ltd) protocol. PCR product containing target gene and protein expression components (including 5' T7 promoter, ribosome binding site, start codon and 3'T7 terminator) was added into cell cell-free synthesis reaction as template. The reactions were carried out at 30 °C for 12 h. Samples were analyzed by SDS-PAGE.

## Far-ultraviolet circular dichroism (Far-UV CD)

The Far-ultraviolet circular dichroism measurements were performed on 0.2 mg/ml, corresponding to 3.50 μM for the TRPM8

MHR1-3 domain and 5.79 μM for the TRPV2 N-terminal ankyrin-repeat like domain (ARD), in a buffer containing 20 mM Hepes (pH 8.0), 150 mM NaCl, and 10% glycerol. The CD spectra were obtained by a circular dichroism spectrometer (Chirascan V100) with a path length cell of 1.0 mm. All experiments were measured from the range of 200 to 250 nm, and scanned with an increase of 2 °C in the range of 10 to 50 °C. The unit of the CD value is converted into the mean residue ellipticity with the equation:

$$[\theta]_{MR} = 100 \times \frac{\theta}{C \times N \times l}$$

where $[\theta]_{MR}$ is the mean residue ellipticity, θ is the ellipticity in millidegrees, C is he concentration of protein in molarity (M), N is the number of amino acid residues of protein, and l is the path length in centimeters.

## Temperature-dependent intrinsic tryptophan fluorescence

Fluorescence emission spectra were measured on a Spectro-fluorometer using 295 nm excitation to reduce the contribution of tyrosine residues. All the samples were prepared in a 0.2 mg/ml, corresponding to 3.50 μM for the TRPM8 MHR1-3 domain and 5.79 μM for the TRPV2 N-terminal ankyrin-repeat like domain (ARD), in a buffer containing 20 mM Hepes (pH 8.0), 150 mM NaCl and 10% glycerol. The temperature was controlled over a range of 10–30 °C in 2 °C increase by using a water circulation system.

## Preparation of mouse TRPM8 cryo-EM structure for MD simulation

To complete missing parts in the cryo-EM structure of mouse TRPM8 in the apo state, this initial model was reconstructed based on SWISS-MODEL (Waterhouse et al, 2018). Four phosphatidy-linositol lipids were added to previously proposed lipid-binding sites (Yin et al, 2019). The initial model was fitted into the cryo-EM density map using the MDFF method (Trabuco et al, 2008). CHARMM36m force field (Huang et al, 2017) was used. The initial model was first optimized with a 1000-step energy minimization using the conjugate gradient with line-search algorithm. Then the system was simulated for 10 ps. The simulation temperature was maintained at 300 K using the Langevin algorithm (Zwanzig, 1973) with a damping coefficient of 1 ps⁻¹. A scaling coefficient of 0.4 was used to reduce the forces derived from the MDFF grid potential. Chirality restraints, cis-peptide restraints, and secondary structure restraints were applied to the protein during MDFF fitting. Generalized-Born implicit solvent model (Ghosh et al, 1998; Tsui and Case, 2000) was used to describe the solvation effects. A time step size of 0.2 fs was used. For nonbonded interactions, a cutoff distance of 16 Å and a switching distance of 15 Å was used to reduce the computational cost.

## MD simulation setup

With the MDFF refined model, we used the Membrane Builder function (Jo et al, 2007) of the CHARMM-GUI webserver (Jo et al, 2008; Lee et al, 2016) to embed TRPM8 model in a 180 Å × 180 Å

bilayer of 1-palmitoyl-2-oleoyl phosphatidylcholine (POPC) lipids, the lower leaf was mixed with 10% PI(4,5)P$_2$, surrounded by a box of water and ions with a buffer distance of 15 Å. To ensure a 0.15 M ionic concentration and zero net charge, 514 Na$^+$ and 322 Cl$^-$ ions were added. The aforementioned 4 PI(4,5)P$_2$ molecules were kept. OpenMM7.5.1 (Eastman et al, 2017) and CHARMM36m force field (Huang et al, 2017) was used to perform MD simulation on a single V100 graphics processing unit (GPU). After energy minimization, six steps of equilibration were performed (with gradually reduced harmonic restraints applied to protein, lipids, water, and ions). Finally, we conducted production MD runs in the NPT ensemble for 1000 ns and 500 ns at 4 and 30 °C, respectively, four replicas were performed for each temperature. MonteCarloMembraneBarostat in OpenMM was applied at 1 bar with an update frequency of every 100 steps. The particle mesh Ewald method (Darden et al, 1993) was used for electrostatics calculations. Analysis and visualization was performed using VMD (Humphrey et al, 1996).

## RMSF analysis

To assess the flexibility of TRPM8 at individual residue positions during our MD simulation, we calculated the root mean square fluctuation (RMSF) as follows: first, we saved 1600 snapshots from first 500-ns MD trajectories of all low and high temperature simulation respectively (with the first 100 ns of each trajectory discarded) to build the low temperature and the high temperature ensembles; second, we superimposed the Cα coordinates onto the initial structure with a minimal root mean square deviation (RMSD); finally, we calculated the following RMSF at residue position $n$ within the range: $\mathrm{RMSF}_n = \sqrt{\frac{1}{M}\sum_{m=1}^{M} |\vec{r}_{mn} - \langle\vec{r}_n\rangle|^2}$,

where $\vec{r}_{mn}$ is the Cα position of residue $n$ in snapshot $m$, $\langle\vec{r}_n\rangle = \frac{1}{M}\sum_{m=1}^{M}\vec{r}_{mn}$ is the average Cα position of residue $n$, and $M$ is the total number of snapshots in each ensemble. We then calculated the average of $\mathrm{RMSF}_n$ for four equivalent residue positions $n$ of the tetramer as the final RMSF for each residue.

## R$_g$ analysis of inward/outward motions

To probe inward/outward motions in TRPM8 at the residue level of details, we used the *measure* command of the VMD program (Humphrey et al, 1996) to calculate the radius of gyration (R$_g$) based on the Cα atoms. Since the tetrameric channel was aligned with respect to the Z-axis, the R$_g$ essentially describes the average distance between each site to the center of the channel in the X-Y plane. Based on the equilibrium MD simulations of the low and high temperature (between 100 and 500 ns), we averaged R$_g$ over four MD trajectories, and then computed the change of average R$_g$ (n) from $T = 30$ to 4 °C to quantify the cold-activated expansion/contraction at residue position n.

## Energetic analysis of nonbonded interaction energy

We used the NAMD (Phillips et al, 2005) Energy plugin of the VMD program (Humphrey et al, 1996) to calculate nonbonded energy [including van der Waals (vdW) and electrostatic energy] in the TRPM8-water-membrane system. A 10-Å switching distance and a 12-Å cutoff distance were used for the nonbonded interactions. The CHARMM36m force field was used for the nonbonded parameters.

## Animals

W137H-*trpm8* mice in the C57BL/6J background were commissioned by Cyagen Biosciences, Inc. (Hangzhou, China). All animal-related experimental procedures were under the guidelines of the Animal Care and Use Committee of the animal facility at Northeast Forestry University, under approval number 2024101. We used adult male C576J strain mice at the age ranging from 8 to 16 weeks for behavioral tests. WT c576j strain mice were used in the control groups. All mice were kept in standard conditions (a 12-h light/dark cycle under standard food and water supplies). All possible efforts were made to reduce the sample size of mice and also to minimize mice suffering. Animals were randomly assigned to groups, and data analysis was performed blinded. Normality was assessed using the Shapiro–Wilk test; if data were normally distributed, Student's *t*-test were used. Data were presented as mean ± standard deviation. DRG neurons were obtained from both male and female mice.

## RNA-sequencing

Total RNA was extracted from the dorsal root ganglion of both wildtype and W137H mutant mice by using the RNeasy Mini Kit (Qiagen). The sequencing data was filtered with SOAPnuke (v1.5.2) by (1) Removing reads containing sequencing adapter; (2) Removing reads whose low-quality base ratio (base quality less than or equal to 5) is more than 20%; (3) Removing reads whose unknown base ("N" base) ratio is more than 5%, afterwards clean reads were obtained and stored in FASTQ format. The clean reads were mapped to the reference genome using HISAT2 (v2.0.4). Bowtie2 (v2.2.5) was applied to align the clean reads to the reference coding gene set, then the expression level of the gene was calculated by RSEM (v1.2.12). The heatmap was drawn by pheatmap (v1.0.8).

## Animal behavior assay

For the two-temperature choice assay, mice were individually confined in two adjacent Plexiglas chambers and allowed to move freely between these chambers. The plate temperature of one chamber was held at 30 °C, and the other ranged from 5 to 50 °C. The time spent by each mouse on the two chambers was recorded every 3 min.

For the temperature preference assay, mice were individually confined in a Plexiglas chamber on a gradient cooling plate ranging from 5 to 50 °C. The movements of mice were recorded by a thermal imaging camera (FLIR T640). The time spent on the plate at different temperatures was analyzed by the FLIR tools.

## Statistics

All experiments have been independently repeated at least three times. All statistical data are given as mean ± s.e.m.

## Data availability

The RNA sequencing data of mouse trigeminal ganglia have been deposited in the NCBI Sequence Read Archive (SRA) under the accession number PRJNA950563 and are available at: https://www.ncbi.nlm.nih.gov/sra/?term=PRJNA950563. The mass spectrometry proteomics data have been deposited in the ProteomeXchange Consortium via the iProX partner repository and are available at: https://www.iprox.cn/page/PSV023.html?url=1634831300401jcmF.

The source data of this paper are collected in the following database record: biostudies:S-SCDT-10_1038-S44319-025-00630-2.

## Peer review information

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

## Acknowledgements

We are grateful to our lab members for their assistance and discussion. We would like to thank Dr. Cheng Ma (Protein Facility of the Core Facility, Zhejiang University School of Medicine) for constructive advice on protein purification. We thank Alibaba Cloud for providing computational resources. This study was supported by the National Natural Science Foundation of China (32571328, 32421001, 32122040, 31971040, and 31800990 to FY; 81972492 and 21904107 to TG), Zhejiang Provincial Natural Science Foundation of China (LRG25C050001 and LR20C050002 to FY; LR19C050001 to TG), and Hangzhou Agriculture and Society Advancement Program (20190101A04 to TG).

## Author contributions

**Lizhen Xu**: Conceptualization; Data curation; Formal analysis; Supervision; Validation; Investigation; Visualization; Methodology; Writing—original draft; Writing—review and editing. **Xiao Liang**: Data curation; Formal analysis; Visualization; Writing—original draft. **Yunfei Wang**: Data curation; Formal analysis. **Han Wen**: Data curation; Formal analysis. **Wenxuan Zhen**: Data curation; Formal analysis; Validation. **Zhangzhi Xue**: Formal analysis. **Fangfei Zhang**: Data curation; Supervision. **Xiao Yi**: Formal analysis. **Xiaoying Chen**: Investigation. **Lidan Hu**: Methodology. **Bei Li**: Resources; Formal analysis. **Bing Zhang**: Resources; Formal analysis. **Zhenfeng Deng**: Formal analysis. **Wei Yang**: Resources; Supervision. **Shilong Yang**: Resources; Supervision; Methodology. **Tiannan Guo**: Resources; Data curation; Supervision; Funding acquisition; Writing—original draft. **Yi Zhu**: Resources; Data curation; Supervision; Funding acquisition; Writing—original draft. **Fan Yang**: Conceptualization; Resources; Data curation; Formal analysis; Supervision; Funding acquisition; Validation; Investigation; Visualization; Methodology; Writing—original draft; Project administration; Writing—review and editing.

Source data underlying figure panels in this paper may have individual authorship assigned. Where available, figure panel/source data authorship is listed in the following database record: biostudies:S-SCDT-10_1038-S44319-025-00630-2.

## Disclosure and competing interests statement

YZ and TG hold shares of Westlake Omics Inc.

## Declaration of generative AI and AI-assisted technologies in the writing process

We acknowledge the use of ChatGPT (OpenAI) for assistance in improving the language and clarity of this manuscript.

# Expanded View Figures

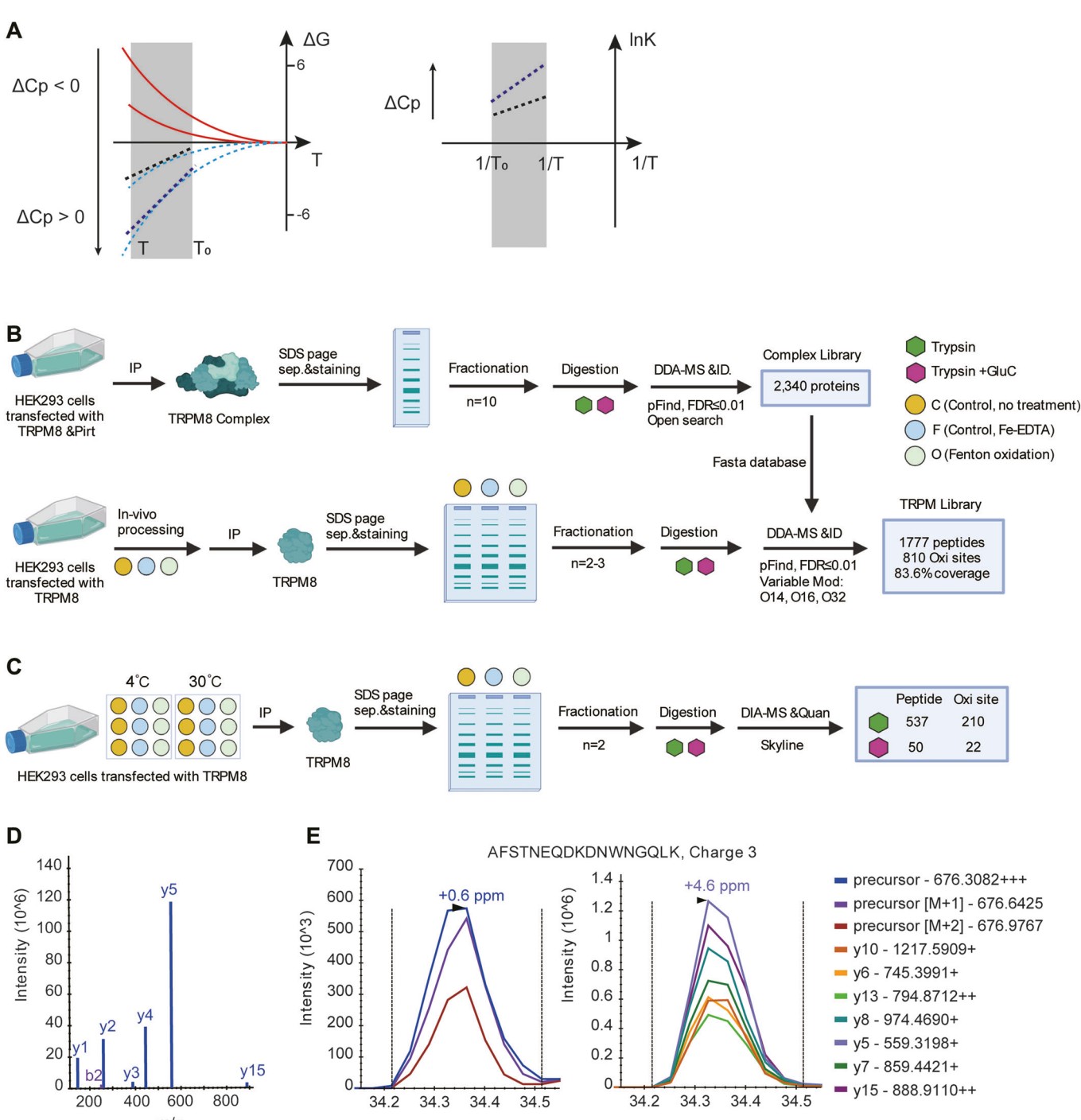

**Figure EV1. Thermodynamic analysis and spectral library generation for TRPM8 oxidation (Related to Fig. 1).**

(A) Residues contributing to ΔCp were mutated to be either polar or hydrophobic. The magnitude of ΔCp influences the degree of curvature, with larger absolute values producing greater curvature. The van't Hoff equation was used to determine ΔH and ΔS from a linear fit of lnK versus 1/T. As the absolute value of ΔCp increases, the slope of the linear fit becomes steeper, resulting in higher absolute values of ΔH and ΔS (gray region). (B, C) The workflow for TRPM8 spectral library generation and oxidation site determination and quantification. DIA data-independent acquisition. (D) Peptide fragmentation spectra from the spectral library. (E) Extracted ion chromatogram (XIC) groups (left: MS1, right: MS2) from the quantification results of an example peptide AFSTNEQDKDNWNGQLK.

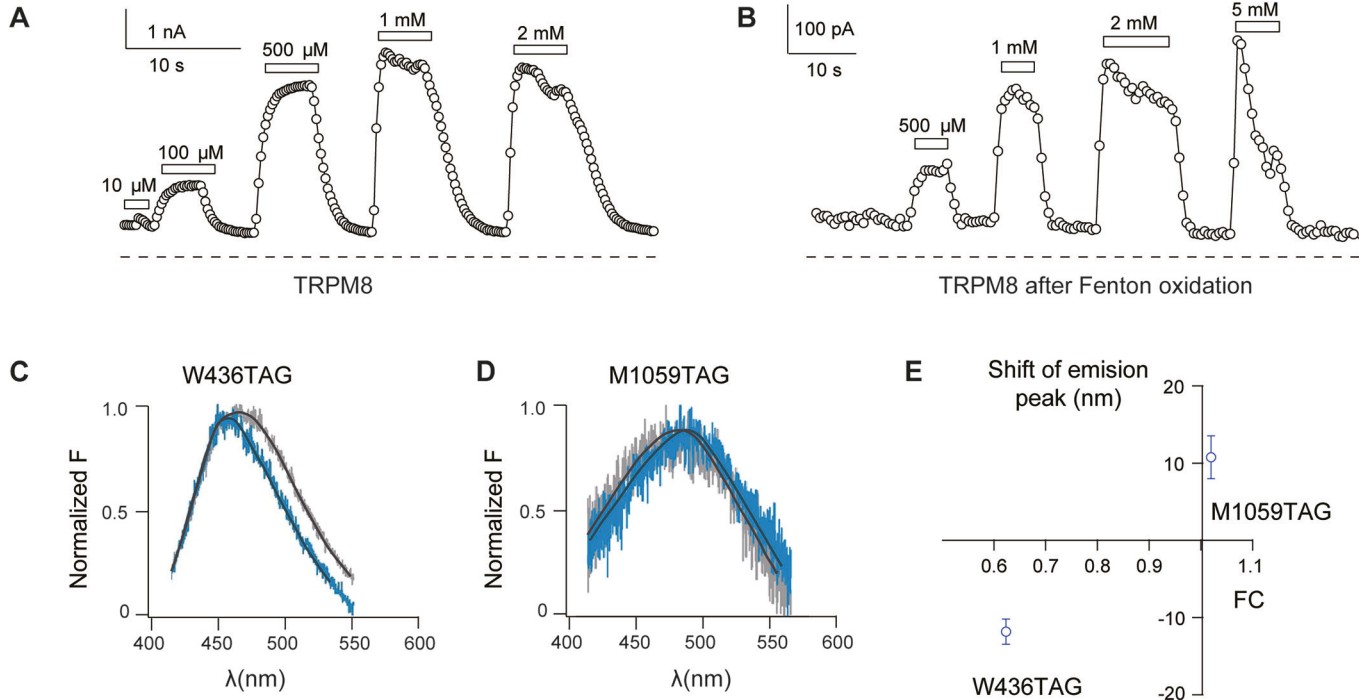

**Figure EV2. Effects of oxidation on TRPM8 channel activity and the properties of ANAP-incorporated mutants.**

(Related to Fig. 1). (**A, B**) Representative whole-cell current recordings of the TRPM8 channel activated by menthol before and after Fenton oxidation. (**C, D**) Representative emission spectra of ANAP incorporated at residue W436 and M1059, respectively. Emission spectra in gray and blue were measured at either 30 or 4 °C, respectively. (**E**) Comparison of shifts in emission spectra peak of ANAP incorporated at residue W436 and M1059 with their FC values measured from HRF-MS. (The y-axis represents the emission spectra peak of ANAP; $n = 3$ biological replicates; data were presented as mean ± s.e.m.).

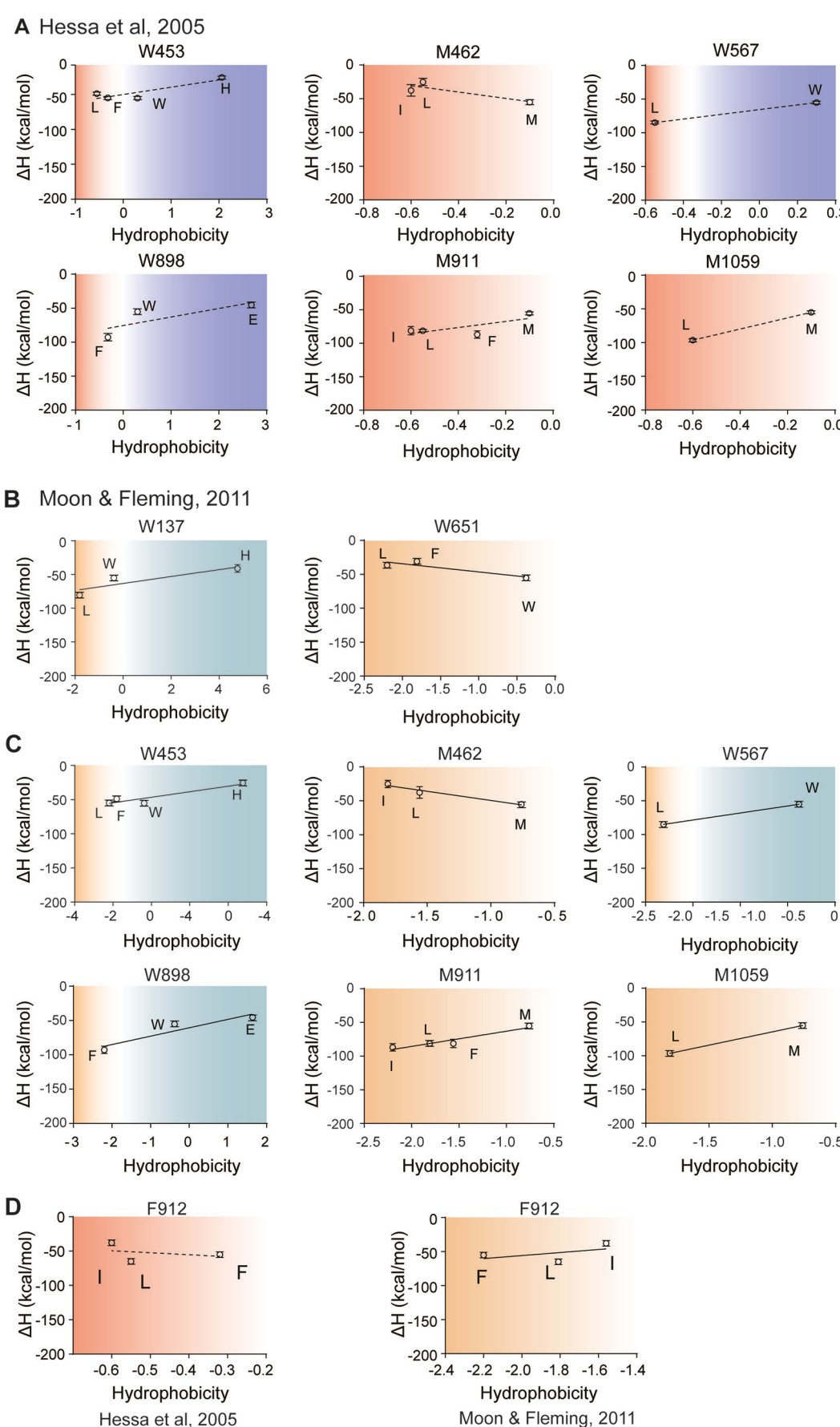

**Figure EV3. Correlation of residue SCH and ΔH in TRPM8 under various SCH hydrophobicity scales during cold activation.**

(Related to Fig. 2). (**A–C**) Correlation between SCH and ΔH values for TRPM8 residues with changes in FC values during cold activation. The hydrophobicity values shown on the x-axis in (**A**) were based on the SCH hydrophobicity scale reported by Hessa et al, whereas those in Panels (**B**) and (**C**) were based on the scale reported by Moon et al. (**D**) The F912 mutation was excluded due to substantial differences in the hydrophobicity scales reported by Hessa et al and Moon et al.

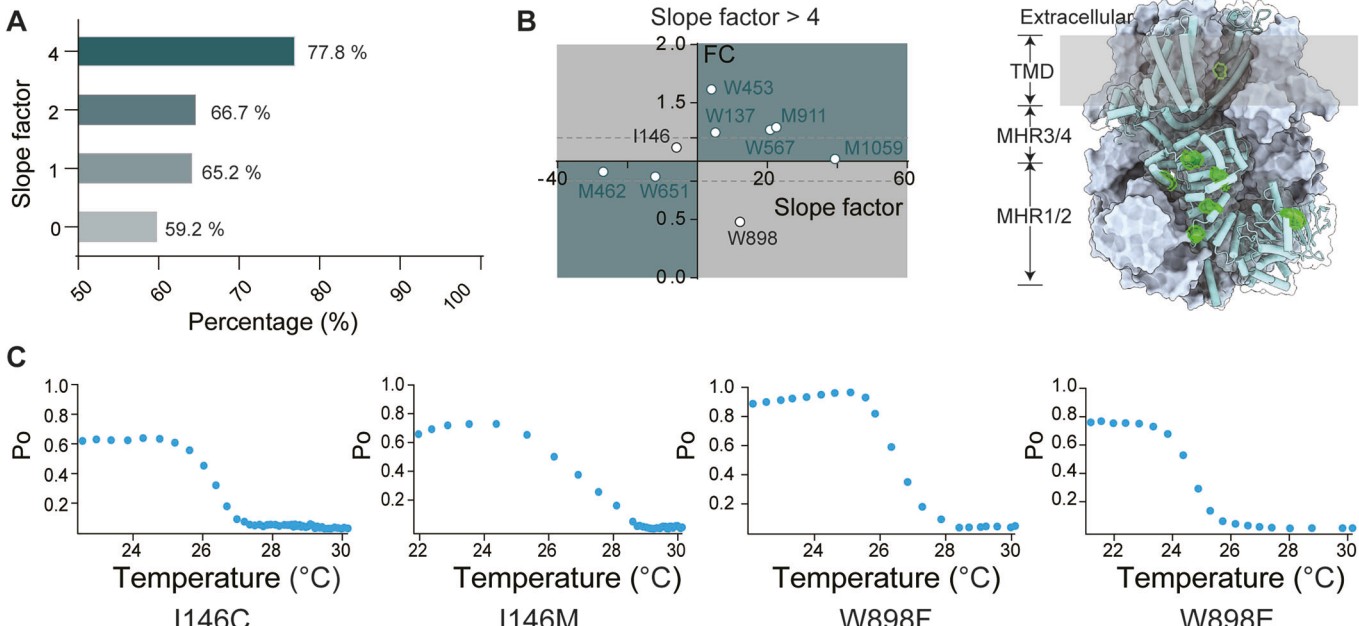

**Figure EV4. Relationship between residue SCH and ΔH in TRPM8 cold sensitivity.**

(Related to Fig. 2). (A) The proportion that aligns with the temperature sensitivity hypothesis increased as the slope factor increased. The slope factor was calculated by fitting SCH and ΔH values to a linear function for each site with buried/exposed changes. The hydrophobicity scale of SCH was determined by Moon et al. The data were then grouped into five classes, with absolute slope factor values greater than 0, 1, 2, and 4, respectively. The x-axis represents the proportion that aligns with the temperature sensitivity hypothesis. (B) Correlation between SCH and ΔH values for TRPM8 residues with changes in FC values during cold activation. The hydrophobicity scale of SCH was determined by Moon et al. The slope factor was plotted against the FC value of the corresponding site. The first and third quadrant, where the sites followed the predictions from the water–protein interaction hypothesis of cold sensing, were shaded in green. The second and fourth quadrant, where the sites did not follow the predictions from the water–protein interaction hypothesis of cold sensing, were shaded in gray. The sites located in the first and third quadrant were mapped onto the cryo-EM structure of TRPM8 with their sidechains shown in green. (C) Representative whole-cell current recordings of TRPM8 mutants that deviated from the hypothesis exhibited cold-activated properties.

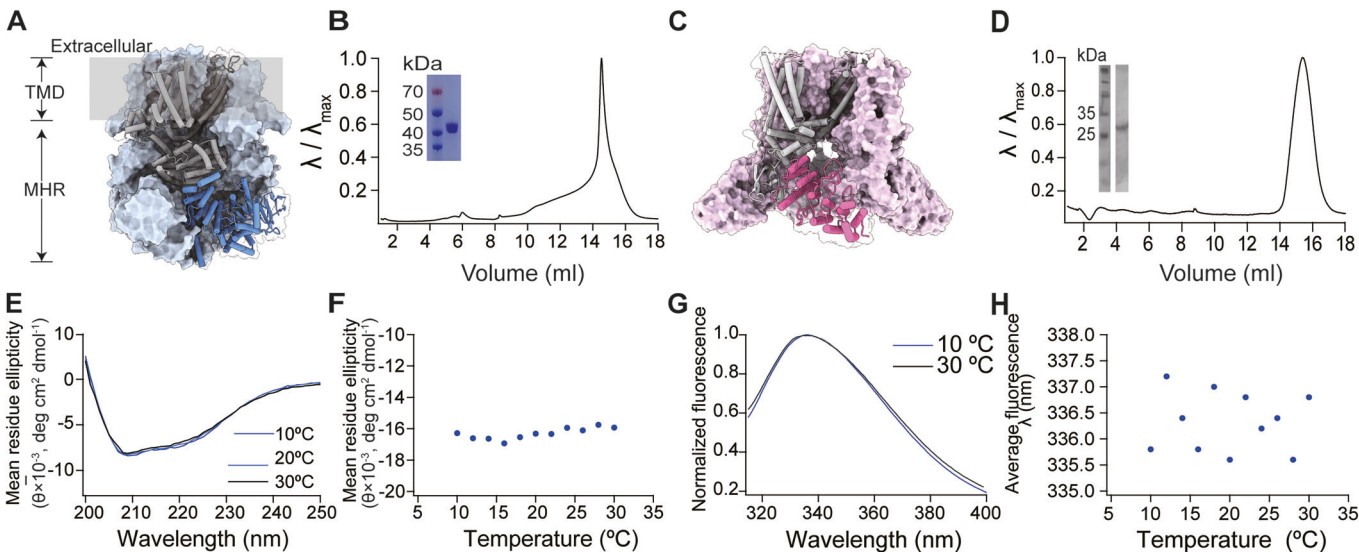

**Figure EV5. Structural and biophysical characterization of TRPM8 MHR1-3 and TRPV2 ARD domains.**

(Related to Fig. 3). **(A)** The location of MHR1-3 domains (colored in blue) in TRPM8. **(B)** Size-exclusion chromatography of the protein of MHR1-3 domains on Superose 6 (GE Healthcare) and SDS-PAGE. **(C)** The location of ARD domains (colored in pink) in TRPV2. **(D)** Size-exclusion chromatography of the protein of ARD on Superose 6 (GE Healthcare) and SDS-PAGE. **(E)** Representative CD spectra of ARD in TRPV2. **(F)** The temperature dependence of the CD spectra. **(G, H)** Representative intrinsic tryptophan emission spectra of ARD in TRPV2 and temperature dependence of intrinsic tryptophan emission peaks, respectively.

