## [Peer Review File · EMBO Reports]

Rational tuning of temperature sensitivity in TRPM8 channel

Lizhen Xu, Xiao Liang, Yunfei Wang, Han Wen, Wenxuan Zhen, Zhangzhi Xue, Fangfei Zhang, Xiao Yi, Xiaoying Chen, Lidan Hu, Bei Li, Bing Zhang, Zhenfeng Deng, Wei Yang, Shilong Yang, Tiannan Guo, Yi Zhu, and Fan Yang

Corresponding author(s): Fan Yang (fanyanga@zju.edu.cn) , Tiannan Guo (guotiannan@westlake.edu.cn), Yi Zhu (zhuyi@westlake.edu.cn), Shilong Yang (syang2020@nefu.edu.cn)

Review Timeline:

Submission Date:	5th Mar 25
Editorial Decision:	19th May 25
Revision Received:	8th Aug 25
Editorial Decision:	26th Sep 25
Revision Received:	9th Oct 25
Accepted:	21st Oct 25

Transaction Report:

Dear Prof. Yang

Thank you for the submission of your research manuscript to our journal. We have now received the full set of referee reports that is copied below.

As you will see, the referees acknowledge that the findings are potentially interesting, but they also raise a number of significant concerns. I have discussed the concerns further with referee #1 and #2 and given that many of these might be addressable with further clarification of methods, data presentation and by providing raw data, I note one important concern from referee 1, i.e., the weak correlation between the slope of hydrophobicity vs ΔH and temperature sensitivity (Fig. 2F). This concern questions the key conclusions of your study.

Given the potential interest of the findings, I would be willing to give you a chance to respond to the referee concerns and revise your study for EMBO Reports. I must however point out, that the revision would need to address all referee concerns and provide strong support for the proposed hypothesis that the hydrophobicity of a given residue correlates with cold sensitivity. If you are willing to engage in such a revision, then please address all referee concerns in full. If you feel that such a revision might be unproductive, as the correlation might remain weak, please contact me. I am happy to discuss the exact revision requirements and next steps further, either by e-mail or in a video chat.

If you decide to embark on the revision, please address all referee concerns in a complete point-by-point response. Acceptance of the manuscript will depend on a positive outcome of a second round of review. It is EMBO Reports policy to allow a single round of revision only and acceptance or rejection of the manuscript will therefore depend on the completeness of your responses included in the next, final version of the manuscript.

We realize that it is difficult to revise to a specific deadline. In the interest of protecting the conceptual advance provided by the work, we recommend a revision within 3 months (August 19th). Please discuss the revision progress ahead of this time with the editor if you require more time to complete the revisions.

As stated above, I am also happy to discuss the revision further via e-mail or a video call, if you wish.

=====
IMPORTANT NOTE:

We perform an initial quality control of all revised manuscripts before re-review. Your manuscript will FAIL this control and the handling will be delayed IN CASE the following APPLIES:

- 1) A data availability section providing access to data deposited in public databases is missing. If you have not deposited any data, please add a sentence to the data availability section that explains that.
- 2) Your manuscript contains statistics and error bars based on $n=2$. Please use scatter blots in these cases. No statistics should be calculated if $n=2$.

=====

- 1) a .docx formatted version of the manuscript text (including legends for main figures, EV figures and tables). Please make sure that the changes are highlighted to be clearly visible.
- 2) individual production quality figure files as .eps, .tif, .jpg (one file per figure). Please download our Figure Preparation Guidelines (figure preparation pdf) from our Author Guidelines pages <https://www.embopress.org/page/journal/14693178/authorguide> for more info on how to prepare your figures.

4) a complete author checklist, which you can download from our author guidelines (<<https://www.embopress.org/page/journal/14693178/authorguide>>). Please insert information in the checklist that is also reflected in the manuscript. The completed author checklist will also be part of the RPF.

5) Please note that all corresponding authors are required to supply an ORCID ID for their name upon submission of a revised manuscript (<<https://orcid.org/>>). Please find instructions on how to link your ORCID ID to your account in our manuscript tracking system in our Author guidelines (<<https://www.embopress.org/page/journal/14693178/authorguide#authorshipguidelines>>)

6) We replaced Supplementary Information with Expanded View (EV) Figures and Tables that are collapsible/expandable online. A maximum of 5 EV Figures can be typeset. EV Figures should be cited as 'Figure EV1, Figure EV2' etc... in the text and their respective legends should be included in the main text after the legends of regular figures.

7) Please include a dedicated "Data Availability" section at the end of the Methods (suggested wording: "The [structural coordinates | microarray | mass spectrometry] data from this publication have been deposited to the [name of the database] database [URL] and assigned the identifier [accession | permalink | hashtag]."). Should this not apply, this should still be stated as "This study includes no data deposited in external repositories."

Additional information on source data and instruction on how to label the files are available <<https://www.embopress.org/page/journal/14693178/authorguide#sourcedata>>

10) Figure legends and data quantification:
The following points must be specified in each figure legend:

- the name of the statistical test used to generate error bars and P values,
 - the EXACT p-values,
 - the number (n) of independent experiments (please specify technical or biological replicates) underlying each data point,
 - the nature of the bars and error bars (s.d., s.e.m.)
- If the data are obtained from n {less than or equal to} 5, show the individual data points in addition to the SD or SEM.
- If the data are obtained from n {less than or equal to} 2, use scatter blots showing the individual data points.

See also the guidelines for figure legend preparation:
<https://www.embopress.org/page/journal/14693178/authorguide#figureformat>

11) Our journal encourages inclusion of *data citations in the reference list* to directly cite datasets that were re-used and obtained from public databases. Data citations in the article text are distinct from normal bibliographical citations and should directly link to the database records from which the data can be accessed. In the main text, data citations are formatted as

follows: "Data ref: Smith et al, 2001" or "Data ref: NCBI Sequence Read Archive PRJNA342805, 2017". In the Reference list, data citations must be labeled with "[DATASET]". A data reference must provide the database name, accession number/identifiers and a resolvable link to the landing page from which the data can be accessed at the end of the reference. Further instructions are available at <<https://www.embopress.org/page/journal/14693178/authorguide#referencesformat>>.

12) All Materials and Methods need to be described in the main text using our 'Structured Methods' format. According to this format, the Methods section includes a Reagents and Tools Table (listing key reagents, experimental models, software and relevant equipment and including their sources and relevant identifiers) followed by a Methods and Protocols section describing the methods, ideally using a step-by-step protocol format. The aim is to facilitate adoption of the methodologies across labs. Please download and fill our Reagents and Tools Table template (.docx), which you can find in our author guidelines:

13) As part of the EMBO publication's Transparent Editorial Process, EMBO Reports publishes online a Review Process File to accompany accepted manuscripts. This File will be published in conjunction with your paper and will include the referee reports, your point-by-point response and all pertinent correspondence relating to the manuscript.

Yours sincerely,

=====

Referee #1:

The manuscript by Xu et al studies the mechanism of cold activation of the TRPM8 ion channel. They identify groups of clustered residues that undergo conformational rearrangements between buried and exposed states during cold activation by hydroxyl radical footprinting mass spectrometry (HRF-MS). Then they perform extensive mutagenesis of these residues to more hydrophobic, or less hydrophobic residues and test cold activation of the mutants. They calculate enthalpy changes from Van't Hoff plots, and correlate the hydrophobicity of the given residue with those changes to calculate the slope factor. Then they plot the fold changes from HRF-MS experiments as a function of the slope factors. They conclude that residues that showed an increase in oxidation efficiency upon cooling to 4C showed an increase in cold sensitivity (slope factor) with increased hydrophobicity at that position. Vice versa, residues that showed a decrease in oxidation efficiency upon cooling to 4C showed a decrease in cold sensitivity (slope factor) with increased hydrophobicity at that position.

The manuscript uses an innovative techniques to tests a key hypothesis in temperature activation using a variety of approaches, and the authors generated an enormous amount of data. Unfortunately, at the end, the key data give a quite weak support to the tested hypothesis, given the very loose correlation between the slope of hydrophobicity vs delta-H and temperature sensitivity. Also, the experiments are described in a very cursory manner and only highly processed data are shown which makes evaluating the data very difficult. Overall, I cannot support the publication of the manuscript in EMBO Reports. See more detailed comments below.

1. The correlation between the slope factor and FC is quite weak. There are positions that behave as expected by the hypothesis, but there are plenty of positions that do not, depending on the threshold of what is plotted. This is in stark contrast to the almost definitive language in the abstract and the title.

The authors use distribution in quadrants (Fig. 2F & Fig. S4B) to evaluate the data, and present the overall agreement of this

quadrant distribution with their hypothesis (Fig. 2E) using various thresholds for the slope factor. I find this problematic, as some data points that do not support the hypothesis are used as supporting points, i.e. M1059 is placed in the upper right quadrant and used as a supporting data point, but to me this is a questionable interpretation, as this residue is more of a negative control given the FC of close to one (1.02), which means its exposure does not change upon cold activation, yet it has the largest slope factor by a wide margin. Also, this method does not take into account the amplitude of the effects. Additional issues: why is residue F912 not plotted in Fig. 2F? It seems to give strong support if they look at hydrophobicity by the Hessa method, but it gives an opposite effect by the Moon method. Without seeing the actual data, it is hard to tell how this is possible. I would be wondering if there is any significant correlation if they performed a linear regression, which would take into account the amplitude of the effects also.

2. The manuscript is written in a confusing manner, difficult to read, and it is really hard to decipher what experiments were performed and how. For example: Table S3 lists 212 mutations, but experimental data is only shown for 16 mutants plus wild type TRPM8 (Fig. S3). The tables must be based on more data, but it is still impossible to tell which mutants were tested and which ones were not. For W137 it is disclosed that many mutants are non-functional or barely functions. Is this also the case for mutations of the other residues which are not shown in figures? Even the most comprehensive figure with the mutants (Fig. S3) have less mutants for most positions than what the the list of primers (Table S3) shows.

3. Along the same lines: There is not a single representative patch clamp experiment shown which would disclose how the actual experiments were performed. In the absence of that, it is difficult, if not impossible to understand what was done. For example, the methods state that measurements were performed at +/- 80 mV, but whether that means a ramp protocol or a step protocol is unclear, and whether the data were calculated from the currents at -80 or +80 mV is also unclear. This is important, because TRPM8 has a weak voltage dependence, and voltage influences cold activation and vice versa.

4. The legend for figure 1 shows that some of the cells were transfected with Pirt and TRPM8. There is not a single word in the manuscript explaining what Pirt is, and why it was co-transfected with TRPM8.

5. There are multiple issues with presentation, for example: Figure 1 and Figure S1 are the same, or at least extremely similar, with somewhat different legends. Table 1 and Table 2 shows the same data, but the slope factor calculated with different methods. I would combine them. Some of the "representative" patch clamp experiments ha P_o as the y-axis, but it is not clear how P_o was derived from whole cell patch clamp experiments. Are these currents normalized to the effect of Menthol? At what voltage? In the legend for Figure 2A and C, the middle panel id referred to as Van't Hoff plot, but there is no middle panel and no Van's Hoff plot in the actual figure. The main text refers to Fig. 1 A as a Van's Hoff plot, but those seem to be theoretical curves, and not actual data.

6. All conclusions are made from the enthalpy changes, which I think correlate with the steepness of the cold activation curve. It is clear however that some of the mutants also affected other parameters, such as temperature threshold, which in some cases went the opposite direction, see Fig 2A for the W137L mutant, which is mentioned as having higher cold sensitivity, but it actually requires colder temperatures to be activated than WT. The authors should provide data on all the different effects the mutants had on channel activity with proper statistics and discussion. At the minimum they should plot, or include in a table, mean menthol-induced and cold-induced currents (or at least their ratio), and cold thresholds both at -80 and +80 mV.

7. The data with the mouse model is entirely predictable, and it does not add much to the main message of the paper. Of course, if you replace the wild type TRPM8 in mice with a mutant that essentially does not respond to cold, it will affect cold sensitivity in a similar manner that is well documented in mice with genetic deletion of TRPM8. Also, if the authors would like to include these data in a publication, they also need to provide clear description of the effect of the mutation, with proper statistic, as opposed to a not very well-defined representative trace in Fig. 2G. They may also consider characterizing cold-induced and agonist- induced currents or Ca^{2+} signals in DRG neurons prepared from those mice, compared to wild type.

8. Page 6, line 29: "We observed that except for the residue M911, the slope factors for all the other 28 residues followed the same trend (Appendix Table S1-2)." Did you mean W898? Also, did you meant to cite Table 1 and 2?

9. Appendix Fig S2C shows a clear shift in the spectrum of ANAP, but panel D shows two overlapping noisy traces, and the text and the legend (E) claims a similar shift in emission in the opposite direction.

MINOR:

Title: either say TRPM8 channels, or the TRPM8 channel or TRPM8. Same applies to the abstract

Fig.2E should come after Fig2F, as it is impossible to understand what is plotted there before a thorough look at Fig. 2E.

Referee #2:

The work by Xu and colleagues addressed an important unsolved question in biology, that is, how a small group of ion channels response to temperature changes with much higher sensitivities than a "normal" protein. Many previous attempts at answering

this question have been reported; they were based on mostly indirect measurements hence the observations have limited information content or relevance. The present study borrowed a recently developed footprinting method for contrasting surface residues at different conformations of membrane proteins. It utilizes free radicals to oxidize residues exposed at protein surface; labeled residues are subsequently identified by mass spectroscopy. Conducted at high and low temperatures that would keep the cold-sensing TRPM8 ion channel mostly in its closed or open state, the authors identified a group of residues in the N-terminal intracellular region of the channel. Tested example residues exerted substantial impacts on the channel's temperature sensitivity when mutated, and their exposed-buried transition confirmed by fluorescence recordings. The study is highly significant in that it offers a set of fresh data from a new type of experiment towards understanding the elusive heat-sensing mechanism. Its publication is strongly recommended. Nonetheless, this apparently hastily prepared manuscript contains a number of errors; some are listed below. Additional information that would improve the manuscript is also suggested.

1. When referring to a temperature level, there should be no space between the value and the °C sign (4°C instead of 4 °C). A space is added when the value represents a range of temperatures, e.g., from 20°C to 24°C 4 °C. (In general, there should be a space between a value and its unit, e.g. 5 mL, 5 min, 100 mM mentioned in Methods.)
2. An estimate of the percentage coverage for the total residues changing accessibility would be very informative and key in evaluating the functional role of reported sites (as well as supporting the claim of an "exhaustive" search in the Discussion section). For example, Ref. Liu et al. (2020) suggests that 14 out of the 20 amino acids can be effectively labeled by hydroxyl radicals under their experimental settings. The MS library coverage rate reported in this manuscript was 84%, hinting at incomplete mapping.
3. It is unclear how P_o was determined in this study, and if the method was equally applicable to the wildtype and channels with a mutation at key sites.
4. Figure 1C Y-axis should be in log scale to better show the symmetrical thresholds used. Similarly, other figure panels using ratios would benefit a log transformation.
5. Figure 1 legend: the equation is incorrect (missing ΔS_0^*T).
6. Figure 2F: the structural demonstration of identified sites is not very effective. A structural figure with better 3D perspective and a more easily identifiable color for the residues would be preferred.
7. Figure 3C: colors in the inset are switched.
8. Figure 3E: the label for warm colors should be 30{degree sign}C instead of 4{degree sign}C.
9. Figure 4E: double-check figure legend colors.
10. Tables 4&5, what do the blue boxes signify? Also, Table 5 has no values, so the mentioning of "mean +/- SEM" should be removed.
11. The term "melting temperature" (in referring to Figure 3B) is undefined and potentially misleading.
12. Page 19: should c576j be C57BL/6J?

Referee #3:

This is a well-conceived project and well-written paper on how protein sense temperature. It should have many applications and implications.

1. After reading and editing hundreds of PhD theses and scientific papers, I find that this paper is nearly perfect in grammar and style, which is unheard of for authors who have English as a second or higher language. This perfection suggests an AI method of preparation. I'm not sure this is fair (to those who labor over their work to refine each sentence); at least the use of AI should be acknowledged, as one would for a English editor. Indications are "heuristic hypothesis" (I have rarely heard a non-native speaker use this elegant expression) and that the voice style of the experimental is inconsistent, varying between active and passive voices (apparently AI has not learned that the style for experimental sections is past tense, passive voice).

2. The literature citations are incomplete. The authors cite the pioneering work of Sze, Chance, and a methods paper by Gross. No perspective is given. There are reviews (in Chem Rev, for example) on protein footprinting that would be good to cite. There is pioneering work by Lisa Jones on footprinting in cells and by Josh Sharp. Targeted membrane protein footprinting in cells has seen new method development and application; see, for example: (<https://onlinelibrary.wiley.com/doi/abs/10.1002/anie.202424779>)

Other reactive species (carbenes) have been demonstrated. More perspective is needed.

3. A problem with footprinting by Fenton Chemistry is the reactions are slow, opening the possibility that the early times of footprinting cause the protein to change conformation but now give misleading footprints (ultimately soluble protein precipitate with excessive HRF, showing the problem of excessive labeling). This shortcoming is bypassed with synchrotron or laser activation to limit the reaction time to milliseconds or shorter. This problem has motivated the development of a pulsed discharge lamp to form the radicals (GenNext Technologies) and plasmas (U of Wisc). In the case of this paper, the HRF is used as a "scout" experiment for the design of mutational and fluorescence studies, a nice design. Nevertheless, comments are needed for the excessive labeling problem and to provide perspective on this issue.

4. For concentrations, pls use M, in addition to mg/mL in the fluorescence and other sections.

5. The range for "significant change in the footprinting is FC between 0.83 and 1.2, somewhat arbitrary. This should be defended. Why not use t tests?

6. The coverage of the protein in the digestion step is 83.6%, which is not defended and seems inadequate. What about the remaining 17% of the protein? Could it be informative?

I think these issues can be resolved without doing more experiments.

Dear Editor and Reviewers,

We sincerely appreciate the time and effort you have dedicated to evaluating our manuscript (Manuscript ID: **EMBOR-2025-61472V2**, Title: **Rational tuning of temperature sensitivity in the TRPM8 channel**). We are grateful for the constructive comments, which have helped us improve the quality of our work. In response to the reviewers' suggestions, we have carefully revised the manuscript and addressed all concerns point by point. Additionally, we have reformatted the citations and references to strictly follow the EMBO Reports' style guidelines. Below, we provide a detailed response to each comment, along with the corresponding revisions in the manuscript.

Response to Reviewer #1

We thank the reviewer for their thorough and critical evaluation of our manuscript and for acknowledging the innovative techniques and the substantial dataset we generated. Below we respond to each comment in detail and have revised the manuscript accordingly to address all concerns.

1. The correlation between the slope factor and FC is quite weak. There are positions that behave as expected by the hypothesis, but there are plenty of positions that do not, depending on the threshold of what is plotted. This is in stark contrast to the almost definitive language in the abstract and the title. The authors use distribution in quadrants (Fig. 2F & Fig. S4B) to evaluate the data, and present the overall agreement of this quadrant distribution with their hypothesis (Fig. 2E) using various thresholds for the slope factor. I find this problematic, as some data points that do not support the hypothesis are used as supporting points, i.e. M1059 is placed in the upper right quadrant and used as a supporting data point, but to me this is a questionable interpretation, as this residue is more of a negative control given the FC of close to one (1.02), which means its exposure does not change upon cold activation, yet it has the largest slope factor by a wide margin. Also, this method does not take into account the amplitude of the effects. Additional issues: why is residue F912 not plotted in Fig. 2F? It seems to give strong support if they look at hydrophobicity by the Hessa method, but it gives an opposite effect by the Moon method. Without seeing the actual data, it is hard to tell how this is possible.

I would be wondering if there is any significant correlation if they performed a linear regression, which would take into account the amplitude of the effects also.

Answer: We thank the reviewer for the thoughtful and constructive comments. We agree that the correlation between the slope factor and fold change (FC) from HRF-MS is modest and reflects a trend rather than a strict linear relationship.

However, we believe that such modest correlation reflects the inherent difficulties in accurate determination of thermodynamic properties of ion channel proteins in live cells. For instance, we have re-analyzed data from a previous study of temperature dependence in voltage-gated ion channels (Sandipan Chowdhury et al, Cell, 2014) using similar metrics (Figures R1 and R2). This study also showed a modest statistical correlation as we observed in our experiments. Importantly, when retaining all data points—including outliers—our R^2 values were comparable to those reported (Figure R2, scatter plot on the right). This reinforces the notion that temperature sensitivity is influenced by multiple factors beyond residue exposure alone. These findings further support our interpretation that, although the observed trend is not strictly predictive, it remains mechanistically meaningful.

REDACTED:

Figure 2D, 2E

Figure 3C, 3D

from Chowdhury et al. (2014)
<https://doi.org/10.1016/j.cell.2014.07.026>

Figure R1. Re-analysis of data from Chowdhury et al. (2014)

REDACTED:
Figure 2D, 2E

Figure 3C

from
Chowdhury et al. (2014)
<https://doi.org/10.1016/j.cell.2014.07.026>

Figure R2. Comparable R^2 values between our dataset and Chowdhury et al. (2014) when including all data points.

Moreover, we believe this modest correlation reflects a meaningful scientific phenomenon. As shown in Figure R3, our original data points are not uniformly distributed across the four quadrants, with notably fewer points in the first and third quadrants. This non-uniform distribution pattern suggests directionality in our data. Specifically, changes in hydrophobicity influence ΔH in a consistent manner depending on residue exposure (buried vs. exposed), supporting our hypothesis of a water-protein interaction-based temperature-sensing mechanism. While we do not propose a quantitative model, we aim to highlight this mechanistic tendency as a step forward to understand the temperature sensing in TRP channels. Therefore, in response to the reviewer's comment, we have revised the main text (Page 6, Lines 38-43) to better acknowledge the limitations of the correlation while retaining the biological significance of our observed trend.

Hessa et al, *Nature*, 2005

Moon&Fleming, *PNAS*, 2011

Figure R3. Non-uniform distribution of the correlation between SCH and ΔH values for TRPM8 residues with changes in FC values during cold activation. Our data points were clustered in the first and third quadrant, with few points located in the second or fourth. The hydrophobicity scale of SCH was determined by either Hessa et al. or Moon et al., respectively.

Regarding residue M1059: We appreciate the reviewer's observation that M1059 exhibits a FC value close to 1 (~1.02), suggesting minimal change in surface accessibility upon cold activation. According to our model, not all residues participate equally in temperature sensing. For residues not directly involved in sensing temperature change, it is possible to observe a substantial $\Delta SASA$ but minimal ΔH , or vice versa. Interestingly, M1059 appears to support this view: despite the small FC, its mutation caused a shift in ΔH . Of course, we cannot rule out the possibility that it had an impact on thermodynamic sensitivity independent of solvent exposure.

Moreover, we retained this residue in the analysis because it showed a consistent trend across both hydrophobicity scales (Hessa et al. and Moon et al.), and the corresponding mutant generated robust currents in electrophysiological recordings. In contrast, many other sites produced non-functional mutants, so the inclusion of M1059 provides valuable information for evaluating the broader trend, even if it may be viewed as an outlier.

Regarding residue F912: We tested several mutations at this site, among which only F912I and F912L yielded measurable functional responses. However, the hydrophobicity value of isoleucine differs significantly between the Hessa and Moon scales, leading to inconsistent trends depending on the scale used (see Figure R4). We have revised the main text (Page 7, Lines 5-20) to explain the conflict.

REDACTED:

Two panels from

Hessa, T., Kim, H., Bihlmaier, K. et al. (2005) <https://doi.org/10.1038/nature03216>
and
C.P. Moon, & K.G. Fleming, <https://doi.org/10.1073/pnas.1103979108> (2011)

Figure R4. Isoleucine exhibits different hydrophobicity values under two established hydrophobicity scales.

2. The manuscript is written in a confusing manner, difficult to read, and it is really hard to decipher what experiments were performed and how. For example: Table S3 lists 212 mutations, but experimental data is only shown for 16 mutants plus wild type TRPM8 (Fig. S3). The tables must be based on more data, but it is still impossible to tell which mutants were tested and which ones were not. For W137 it is disclosed that many mutants are non-functional or barely functions. Is this also the case for mutations of the other residues which are not shown in figures ? Even the most comprehensive figure with the mutants (Fig. S3) have less mutants for most positions than what the the list of primers (Table S3) shows.

Answer: We thank the reviewer for the helpful comments and apologize for

the confusion. In total, we generated and tested 186 TRPM8 point mutants, but not all of them yielded functional channels. Importantly, not all mutants yielded functional channels. Many showed no detectable current or only minimal activity in patch-clamp recordings, and therefore could not be included in further thermodynamic analyses. Only mutants that produced functional current in response to menthol or cold stimulation were analyzed and shown in figures such as Fig. 2 and Dataset EV3 (formerly named Table S3).

To address this point more transparently, we have taken the following steps:

1. An additional sheet has been added to Dataset EV3 (formerly named Table S3), indicating whether the mutants was functionally expressed.
2. We have clarified this in the main text (Page 6, Lines 20-23), with the following added statement: only functionally expressed mutants—defined as those capable of producing detectable current in response to menthol or cold—were included in the electrophysiological and thermodynamic analyses. Mutants that failed to yield current or showed ambiguous behavior were excluded.

As the reviewer pointed out, loss of function at the W137 site was also observed at several other mutation sites, particularly those involving changes such as charge reversal. These non-functional mutations have now been clearly indicated in the updated Dataset EV3 (formerly named Table S3). We hope these revisions and clarifications improve the clarity of the manuscript and resolve the reviewer's concern.

3. Along the same lines: There is not a single representative patch clamp experiment shown which would disclose how the actual experiments were performed. In the absence of that, it is difficult, if not impossible to understand what was done. For example, the methods state that measurements were performed at +/- 80 mV, but whether that means a ramp protocol or a step protocol is unclear, and whether the data were calculated from the currents at -80 or +80 mV is also unclear. This is important, because TRPM8 has a weak voltage dependence, and voltage influences cold activation and vice versa.

Answer: We thank the reviewer for the valuable comment. To clarify, we used a voltage-step protocol ranging from -80 mV to +80 mV during patch-clamp recordings. Unless otherwise specified, cold- and menthol-induced currents

were recorded at +80 mV. We chose +80 mV to capture the outward currents relevant for TRPM8 activation. Additionally, the open probability (P_o) was estimated by normalizing the current amplitude under saturating menthol (1 mM) stimulation, assuming maximal channel open probability under these conditions. These details have been added to the Methods section to enhance clarity and reproducibility.

4. The legend for figure 1 shows that some of the cells were transfected with Pirt and TRPM8. There is not a single word in the manuscript explaining what Pirt is, and why it was co-transfected with TRPM8.

Answer: We thank the reviewer for pointing out this omission. Pirt is a membrane protein known to modulate TRPM8 channel activity and expression. We co-transfected Pirt along with TRPM8 to achieve more targeted and precise functional measurements. Additionally, co-expression with Pirt allows the formation of a specific binding partner complex, which helps narrow the search space in our analyses and enables more stringent control of the false discovery rate. We have now included this explanation in Methods to clarify its role in our study.

5. There are multiple issues with presentation, for example: Figure 1 and Figure S1 are the same, or at least extremely similar, with somewhat different legends. Table 1 and Table 2 shows the same data, but the slope factor calculated with different methods. I would combine them. Some of the "representative" patch clamp experiments ha P_o as the y-axis, but it is not clear how P_o was derived from whole cell patch clamp experiments. Are these currents normalized to the effect of Menthol ? At what voltage ? In the legend for Figure 2A and C, the middle panel id referred to as Van't Hoff plot, but there is no middle panel and no Van's Hoff plot in the actual figure. The main text refers to Fig. 1 A as a Van's Hoff plot, but those seem to be theoretical curves, and not actual data.

Answer: We sincerely thank the reviewer for the constructive feedback regarding the presentation of figures and tables. In response, we have made the following revisions and clarifications:

1. Due to previous figure misplacement, Figure S1 is not a duplicate of Figure

1 but rather a supplement and extension. Figure S1 provides a detailed explanation of the theoretical framework of temperature sensing and illustrates the experimental workflow of mass spectrometry, which helps readers better understand the study design. Therefore, this figure holds independent and essential value.

2. Tables 1 and 2 have been combined into a single comprehensive table, with slope factors calculated by both methods presented side-by-side for direct comparison and clearer understanding.

3. Regarding the calculation of P_o (open probability): P_o is defined as the cold-induced current normalized to the menthol-induced current, measured at a holding potential of +80 mV. A detailed description of the calculation method has been added to the Methods section.

4. The descriptions of the Van't Hoff plots in the figure legends of Figures 2A and 2C were residual from earlier versions and were not deleted in time; these have now been removed.

5. We have thoroughly reviewed and corrected all figure numbers, legends, and in-text figure references to ensure accuracy, consistency, and clarity throughout the manuscript.

6. All conclusions are made from the enthalpy changes, which I think correlate with the steepness of the cold activation curve. It is clear however that some of the mutants also affected other parameters, such as temperature threshold, which in some cases went the opposite direction, see Fig 2A for the W137L mutant, which is mentioned as having higher cold sensitivity, but it actually requires colder temperatures to be activated than WT. The authors should provide data on all the different effects the mutants had on channel activity with proper statistics and discussion. At the minimum they should plot, or include in a table, mean menthol-induced and cold-induced currents (or at least their ratio), and cold thresholds both at -80 and +80 mV.

Answer: Thank you for the valuable suggestion. We acknowledge that although our study primarily uses enthalpy changes (ΔH) to assess the steepness of the cold activation curve, some mutants also affect other channel properties, such as the temperature activation threshold. We have added Table 2, which summarizes the cold activation thresholds measured at -80 mV

and +80 mV for mutants of residues highly sensitive to temperature changes, as well as the ratio of cold-induced current to menthol-induced current measured at +80 mV.

7. The data with the mouse model is entirely predictable, and it does not add much to the main message of the paper. Of course, if you replace the wild type TRPM8 in mice with a mutant that essentially does not respond to cold, it will affect cold sensitivity in a similar manner that is well documented in mice with genetic deletion of TRPM8. Also, if the authors would like to include these data in a publication, they also need to provide clear description of the effect of the mutation, with proper statistic, as opposed to a not very well-defined representative trace in Fig. 2G. They may also consider characterizing cold-induced and agonist-induced currents or Ca²⁺ signals in DRG neurons prepared from those mice, compared to wild type.

Answer: We appreciate the reviewer's suggestion. In fact, Figure 4C shows the cold-induced and agonist-induced responses recorded from DRG neurons of the mutant and wild-type mice. Additionally, we have provided the corresponding quantitative analysis of these data in Figure 4D, which clearly demonstrates the effects of the mutation with appropriate statistical support.

8. Page 6, line 29: "We observed that except for the residue M911, the slope factors for all the other 28 residues followed the same trend (Appendix Table S1-2)." Did you mean W898? Also, did you meant to cite Table 1 and 2?

Answer: Thanks a lot for pointing this out. Our original statement was unclear. What we intended to convey is: Although residue M911 exhibited consistent trends under both scoring systems (Table 1), mutations that reduced SCH also abolished channel activation, thereby limiting subsequent investigations and potentially compromising the accuracy of the slope factor. We have also corrected the table citation from the Appendix Tables to Table 1.

9. Appendix Fig S2C shows a clear shift in the spectrum of ANAP, but panel D shows two overlapping noisy traces, and the text and the legend (E) claims a similar shift in emission in the opposite direction.

Answer: Thank you for pointing out this issue. Although the signals in Figure

S2D are noisy and the traces overlap, we performed fitting analysis of the ANAP emission spectra and confirmed a clear spectral shift consistent with that observed in Figure S2C. We have added the fitted curves in the revised manuscript to clarify the correspondence between the description and the data.

Minor Comments:

MINOR:

Title: either say TRPM8 channels, or the TRPM8 channel or TRPM8. Same applies to the abstract

Fig.2E should come after Fig2F, as it is impossible to understand what is plotted there before a thorough look at Fig. 2E.

Answer: In accordance with the suggestion, the terminology referring to 'TRPM8 channel' has been standardized throughout the manuscript.

Figure and table orders have been corrected to follow narrative logic.

Response to Reviewer #2

We thank the reviewer for their positive evaluation of our work and their insightful suggestions. We have carefully revised the manuscript to address all comments. Below, we provide point-by-point responses.

Major Comments:

1. When referring to a temperature level, there should be no space between the value and the oC sign (4oC instead of 4 oC). A space is added when the value represents a range of temperatures, e.g., from 20oC to 24oC □ 4 oC. (In general, there should be a space between a value and its unit, e.g. 5 mL, 5 min, 100 mM mentioned in Methods.)

Answer: We have corrected all instances of temperature formatting throughout the manuscript. For example, "4 oC" is now written as "4°C", and appropriate spacing has been used in expressions involving units (e.g., "5 mL", "5 min", "100 mM").

2. An estimate of the percentage coverage for the total residues changing accessibility would be very informative and key in evaluating the functional role of reported sites (as well as supporting the claim of an "exhaustive" search in the Discussion section). For example, Ref. Liu et al. (2020) suggests that 14 out of the 20 amino acids can be effectively labeled by hydroxyl radicals under their experimental settings. The MS library coverage rate reported in this manuscript was 84%, hinting at incomplete mapping.

Answer: We thank the reviewer for this insightful comment. We fully agree that an estimate of the percentage coverage is critical for evaluating the extent and reliability of our residue accessibility mapping. Based on our mass spectrometry data, we detected ~84% of residues that are theoretically observable under our experimental conditions.

Following the reviewer's suggestion, we now include a dedicated discussion of this point in the revised manuscript (Discussion, Pages 12-13). We acknowledge that our dataset does not provide a complete map of all TRPM8 residues and have replaced references to an "exhaustive" search with terms as "comprehensive" search to avoid overstating our coverage. Nevertheless, we believe that the accessible subset includes many functionally relevant residues and offers a representative view of the conformational dynamics underlying TRPM8 activation. We also explicitly note the potential impact of coverage limitations on interpretation, especially for residues or domains that may be underrepresented due to low labeling efficiency or poor peptide recovery.

3. It is unclear how P_o was determined in this study, and if the method was equally applicable to the wildtype and channels with a mutation at key sites.

Answer: We thank the reviewer for this valuable comment. We have added a detailed description in the Methods section (Page 18) clarifying that P_o (open probability) was estimated by normalizing the current amplitude elicited under saturating menthol stimulation, which is assumed to reflect the maximal

channel open probability. This approach was applied consistently to both wild-type and mutant channels to ensure comparability.

4. Figure 1C Y-axis should be in log scale to better show the symmetrical thresholds used. Similarly, other figure panels using ratios would benefit a log transformation.

Answer: Following the recommendation, we have changed the Y-axis in Figure 1C to a log scale to better visualize symmetric thresholds.

5. Figure 1 legend: the equation is incorrect (missing $\Delta S_0 \cdot T$).

Answer: We thank the reviewer for catching this error. We have corrected the Van't Hoff equation in the legend to include the missing term: $\Delta S_0 \cdot T$.

6. Figure 2F: the structural demonstration of identified sites is not very effective. A structural figure with better 3D perspective and a more easily identifiable color for the residues would be preferred.

Answer: We have replaced the structural representation in Figure 2F with a clearer 3D rendering using Chimera X, with a more intuitive color scheme and a better perspective view to highlight the identified residues.

7. Figure 3C: colors in the inset are switched.

Answer: Corrected. The colors in the inset now match the main panel appropriately.

8. Figure 3E: the label for warm colors should be 30°C instead of 4°C.

Answer: We have corrected the label. The warm color now correctly corresponds to 30°C, not 4°C.

9. Figure 4E: double-check figure legend colors.

Answer: We have double-checked the figure and updated the legend to ensure color labels match the traces correctly.

10. Tables 4&5, what do the blue boxes signify? Also, Table 5 has no values, so the mentioning of "mean +/- SEM" should be removed.

Answer: We now clarify in the legend what the blue boxes indicate. For Table 5, we removed the "mean \pm SEM" label since the table does not contain numerical values.

11. The term "melting temperature" (in referring to Figure 3B) is undefined and potentially misleading.

Answer: We agree this term may be misleading and have now replaced "melting temperature" with "apparent unfolding midpoint temperature" throughout the main text.

12. Page 19: should c576j be C57BL/6J?

Answer: Corrected to "C57BL/6J."

Response to Reviewer #3:

We sincerely thank the reviewer for the thoughtful and constructive feedback, and we are pleased that the reviewer found our study well-conceived and well-written. We have addressed each of your comments below:

1. After reading and editing hundreds of PhD theses and scientific papers, I find that this paper is nearly perfect in grammar and style, which is unheard of for authors who have English as a second or higher language. This perfection suggests an AI method of preparation. I'm not sure this is fair (to those who labor over their work to refine each sentence); at least the use of AI should be acknowledged, as one would for a English editor. Indications are "heuristic hypothesis" (I have rarely heard a non-native speaker use this elegant expression) and that the voice style of the experimental is inconsistent, varying between active and passive voices (apparently AI has not learned that the style for experimental sections is past tense, passive voice).

Answer: We sincerely thank the reviewer for the very generous comments on the grammar and style of our manuscript. The scientific content, structure, and argumentation were developed entirely by the authors. During the writing process, AI-based tools (ChatGPT) were used in a limited capacity to assist with language refinement. We fully understand the reviewer's concerns regarding the fair and responsible use of such tools, and we agree that the use of AI-based tools should be acknowledged—just as a professional language

editor would be disclosed. We have added a statement to the revised manuscript to clarify this point.

In response to the reviewer's helpful observation about inconsistency in the experimental sections, we have carefully revised those parts to ensure consistent use.

2. The literature citations are incomplete. The authors cite the pioneering work of Sze, Chance, and a methods paper by Gross. No perspective is given. There are reviews (in Chem Rev, for example) on protein footprinting that would be good to cite. There is pioneering work by Lisa Jones on footprinting in cells and by Josh Sharp. Targeted membrane protein footprinting in cells has seen new method development and application; see, for example: (<https://onlinelibrary.wiley.com/doi/abs/10.1002/anie.202424779>) Other reactive species (carbenes) have been demonstrated. More perspective is needed.

Answer: We fully agree that adding more background information is important. In the revised manuscript, we have expanded both the Introduction sections to include key review articles and recent primary studies on hydroxyl radical footprinting. Specifically, we now cite the comprehensive reviews, as well as recent advances in in-cell and membrane protein footprinting. These additions provide a broader and more up-to-date contextual framework, thereby strengthening the theoretical foundation of our study.

3. A problem with footprinting by Fenton Chemistry is the reactions are slow, opening the possibility that the early times of footprinting cause the protein to change conformation but now give misleading footprints (ultimately soluble protein precipitate with excessive HRF, showing the problem of excessive labeling). This shortcoming is bypassed with synchrotron or laser activation to limit the reaction time to milliseconds or shorter. This problem has motivated the development of a pulsed discharge lamp to form the radicals (GenNext Technologies) and plasmas (U of Wisc). In the case of this paper, the HRF is used as a "scout" experiment for the design of mutational and fluorescence studies, a nice design. Nevertheless, comments are needed for the excessive labeling problem and to provide perspective on this issue.

Answer: We thank the reviewer for this important comment. We fully acknowledge that Fenton chemistry-based hydroxyl radical footprinting (HRF) has inherent limitations due to its relatively slow reaction kinetics, which may cause conformational changes during labeling, leading to over-labeling and artifacts. In this study, HRF was primarily used as an initial "scouting" tool to screen candidate residues undergoing conformational changes during cold activation. To minimize potential artifacts from labeling, we combined HRF with mutagenesis and fluorescence validation experiments, independently confirming key findings. We also recognize that technological advances

facilitate more precise mechanistic investigations; therefore, we have added a discussion in the revised manuscript addressing the possible over-labeling and conformational perturbations caused by the Fenton method. We further mention advanced techniques such as synchrotron radiation and pulsed radical generation, which offer higher temporal resolution and reduced artifact risks, representing promising approaches for more accurate footprinting in future studies.

4. For concentrations, pls use M, in addition to mg/mL in the fluorescence and other sections.

Answer: We thank the reviewer for pointing this out. We have revised the Methods section accordingly and added both mg/mL and molar concentrations where applicable to improve clarity and reproducibility.

5. The range for "significant change in the footprinting is FC between 0.83 and 1.2, somewhat arbitrary. This should be defended. Why not use t tests?

Answer: Thank you for the reviewer's insightful comment. We have now included a detailed explanation in the Methods section to clarify our rationale for using the fold change (FC) threshold. The selected range of FC between 0.83 and 1.2 (representing a $\pm 20\%$ change) is based on established conventions in previous HRF-MS studies. This threshold takes into account the typical background noise and inter-replicate variability inherent in mass spectrometry measurements, and serves as a practical cutoff to distinguish biologically meaningful conformational changes.

Moreover, compared to statistical tests such as the t-test—which assume normal distribution and require high replicate numbers—the FC-based approach is more robust and conservative in the context of our dataset, which involves low sample numbers and relatively high variability.

6. The coverage of the protein in the digestion step is 83.6%, which is not defended and seems inadequate. What about the remaining 17% of the protein? Could it be informative?

Answer: We thank the reviewer for raising this important point. We have now addressed this issue in the Results and Discussion sections. While a coverage of 83.6% is considered relatively high for membrane proteins and is consistent with previously published HRF-MS studies, we fully acknowledge that the remaining 17% of the sequence may still contain residues critical for conformational regulation. We have explicitly noted this limitation in the revised manuscript to ensure appropriate interpretation of the data.

Dear Prof. Yang

Thank you once more for the submission of your revised manuscript to EMBO reports. I have already informed you about the referee reports we have received. Both referees recommend publication, after a number of minor concerns have been addressed.

I have now completed all editorial checks and list below a few editorial things that we need before we can proceed with the official acceptance of your study.

Please provide a point-by-point response to these in addition to your response to the referees, to speed up our editorial checks.

- Your manuscript will be published in our Reports section. Please combine the Results and Discussion sections and keep the character limit of approx. 27,000 characters in mind (incl. spaces but excluding methods and references).
- The figures should be removed from the manuscript, only the figure legends should remain.
- The Dataset legends should be removed from the manuscript, only remaining in the Excel files.
- Tables 1-5 should be removed from the manuscript as they are not editable and in their current stage formally equivalent to figures. Please upload Tables 1-5 as separate files and please rename them to Table EV1-EV5 with the legends included in each file.
- Please do not use strike-through text to highlight changes in the text. This might interfere with proper type-setting. Yellow text marks are OK.
- The abstract needs to be written in present tense, i.e., the description of your new findings and it does not need to be bolded.
- Sections need to be named and the order should be corrected: Title page - Abstract - Keywords - Introduction - Results - Discussion - Methods - Data Availability - Acknowledgements - Disclosure and Competing Interests Statement - References - Figure Legends - Table(s) - Expanded View Figure Legends.
- According to our editorial authorship guidelines we generally discourage the listing of co-first, or co-corresponding authors in excess of five. Here, we noted that your paper lists 4 co-corresponding authors. While this is still within our recommended limit, we would nevertheless appreciate a short explanation/justification. Thank you very much.
- Figure EV3 refers to Hessa et al and to Moon & Fleming et al. These references were not entirely clear to me, can you please clarify? Do these panels display/reuse data that were already published in these earlier studies?
- The last sentence in the Acknowledgments states: "This work was supported by Alibaba Cloud." Could you please clarify whether this refers to funding, in which case the information needs to be added to the funding information in our online manuscript tracking system. If the Alibaba Cloud is not a funder, then please rephrase.
- Please provide up to 5 keywords on the title page.
- Regarding the Author Contributions, we now use CRediT to specify the contributions of each author in the journal submission system. Therefore, please remove the Author Contributions from the manuscript file and make sure that the author contributions in our online manuscript tracking system are correct and up-to-date. The information you specified in the system will be automatically retrieved and typeset into the article. You can enter additional information in the free text box provided, if you wish.
- We perform a routine check on all supplied quantification source data. In this case I noticed that some "blocks" of numbers have been repeated in the quantification for Figure 2C, left panel. Could you please check the attached file with color-coded duplications and let me know what happened or whether this is justified?
- Please add information on the authority granting mouse work and the reference number for approval in the methods section.
- The Data availability section should exclusively refer to data deposited in public repositories. Therefore, please remove the following statement: "All data needed to evaluate the conclusions of this study are provided in the main text and/or the Supplementary Materials." And "Additional data are available from the corresponding author upon reasonable request."
- Please add URLs in the Data Availability paragraph that resolve directly to the datasets (not just the database).
- The Reagents and Tools table should be removed from the manuscript and only remain uploaded as an individual file. Please

remove the instructions text from the table.

- Please address the following comments regarding Figure Legends (main + EV):

1. Please note that information related to n is missing in the legends of figures 1C, 4D, EV2 E.
2. Please note that the error bars are not defined in the legends of figures 1C, 4D, EV2 E.

- You have a Declaration of AI-assisted writing in the "Acknowledgements". This should be placed as a separate section named "Declaration of generative AI and AI-assisted technologies in the writing process" below the Data Availability section.

- Finally, EMBO Reports papers are accompanied online by

A) a short (1-2 sentences) summary of the findings and their significance,

B) 2-3 bullet points highlighting key results and

C) a schematic summary figure that provides a sketch of the major findings (not a data image).

Please provide the summary figure as a separate file in PNG or JPG format at a size of 550x300-600 pixels (width x height).

Please note that the size is rather small and that text needs to be readable at the final size. Please send us this information along with the revised manuscript.

With kind regards,

=====

Referee #1:

The authors improved the manuscript. Most importantly, I accept their argument that you cannot expect all residues that change surface exposure in response to cold to be functionally involved in cold sensing. Overall, I find the revised manuscript suitable for publication in principle, but there are some minor issues that I would recommend addressing:

1. The finding that the residues mutations in which showed the clearest effects on cold sensitivity are all located in the MHR1-3 domains, and that the isolated MHR1-3 domains showed conformational changes in response to cold, is worth mentioning in the abstract. On the other hand, I feel that the following sentence in the abstract is an overstatement: "the energy changes associated with variations in water-protein interactions were sufficient to trigger cold activation."
2. The authors inserted a vague statement about Pirt without any reference. I would recommend that they provide a more specific description of Pirt, and provide a reference.
3. The methods do not contain a description of the DRG neuron preparation. Please add.
4. The authors added table 2 with listing other electrophysiological properties of the mutants. It only contains 4 residues with a single mutant in each, and no wild type control. Any specific reason for this? I would add additional mutations in these residues, wild type controls, and mutations in the other two residues that behaved accordingly to the overall hypothesis: M911 and W567.
5. There are several figures where t-test was used repeatedly on the same experiment, for example figure 2, figure 4. Analysis of variance would be more appropriate. The effects in Figure 4 are very clear, but in Figure 2 some of the effect may not be significant after correcting for multiple comparisons.

Referee #2:

The authors have adequately addressed all my previous concerns.

Dear Dr. Rembold,

We would like to thank you and the reviewers for your time and constructive feedback on our revised manuscript. We have carefully addressed all editorial and reviewer comments as requested. Below we provide our point-by-point responses.

Responses to Editorial Requests

1. Combining Results and Discussion:

We have combined the Results and Discussion sections into a single “Results and Discussion” section, keeping the total length within approximately 27,000 characters (excluding Methods and References).

2. Figures and Legends:

All figures have been removed from the main text. Only the corresponding figure legends remain in the manuscript.

3. Dataset Legends:

Dataset legends have been removed from the manuscript and retained only in the Excel files.

4. Tables:

Tables 1–5 have been removed from the manuscript and uploaded as separate editable files named Table EV1–EV5, each including its respective legend.

5. Highlighting Changes:

We removed all strike-through text and used yellow highlights to indicate textual changes.

6. Abstract:

The abstract has been rewritten in present tense and is no longer bolded.

7. Section Order:

The manuscript sections have been reordered as requested.

8. Co-corresponding Authors Justification:

The study was a joint effort of four laboratories, each responsible for key aspects including conceptual design and electrophysiological experiments, mass spectrometry analysis, and animal studies. Therefore, four corresponding authors were designated to represent these distinct areas of contribution, which accurately reflects the actual distribution of responsibilities.

9. Clarification on Figure EV3 References:

The parts of the figure referring to *Hessa et al.* and *Moon & Fleming et al.* only cite the hydrophobicity values of amino acids and are used to illustrate principles previously established in cold sensing. This has been clarified in the figure legend.

10. Acknowledgment (Alibaba Cloud):

Alibaba Cloud provided computational resources but no financial support. We have revised the acknowledgment to read:

“We thank Alibaba Cloud for providing computational resources.”

11. Keywords:

Five keywords have been added to the title page.

12. Author Contributions (CRediT):

The Author Contributions section has been removed from the manuscript, and contributions have been updated in the online submission system according to the CRediT taxonomy.

13. Quantification Source Data (Figure 2C):

We have re-examined the data and confirmed that the duplicated blocks were due to a copying error, which has now been corrected and removed.

14. Ethical Approval for Mouse Work:

The authority and approval number have been added to the Methods section: “All animal-related experimental procedures were under the guidelines of the Animal Care and Use Committee of the animal facility at Northeast Forestry University, under approval number 2024101.”

15. Data Availability Section:

We removed the generic statements and now only list datasets deposited in public repositories with direct URLs.

16. Reagents and Tools Table:

The Reagents and Tools table has been removed from the manuscript and uploaded as a separate file without instructional text.

17. Figure Legends (n and Error Bars):

We have added the missing information on sample size (n) and error bar definitions for Figures 1C, 4D, and EV2E.

18. Declaration of AI-assisted Writing:

This statement has been moved to a separate section titled “Declaration of generative AI and AI-assisted technologies in the writing process,” placed immediately below the Data Availability section.

19. Summary Materials (A–C):

We have provided:

A. 2-sentence summary of the findings and their significance

B. 3 bullet points highlighting the key results

C. A schematic summary figure (600 × 300 px, PNG format) as a separate file.

Responses to Referee #1

1. Abstract revision (MHR1–3 domain & energy statement):

We have mentioned in the abstract that the most pronounced cold-sensitive residues are located in the MHR1–3 domains and toned down the statement about water–protein interaction energy change.

2. Pirt reference:

We have expanded the Pirt description and added the appropriate reference.

3. DRG neuron preparation:

The methods section now includes a detailed description of the DRG neuron preparation protocol.

4. Table 2 data completeness:

Table 2 presents the other electrophysiological properties of the cold-sensitive amino acid mutants. The corresponding wild-type controls have been added, and the table has been updated and refined accordingly.

5. Statistical analysis:

We appreciate the reviewer's concern on our statistic methods. In response to this suggestion, we have reanalyzed the data in Figure 2 using one-way ANOVA instead of multiple t-tests. The results confirmed that there are significant differences among the groups, supporting our original conclusions. We thank the reviewer for this valuable suggestion, which has helped us improve the statistical rigor of our analysis.

For Figure 4, we believe that using t-test is more appropriate. When we set the temperature of plate B to a specific value, we just compare the W137H mice versus the WT mice. For instance, when plate B was set to 5°C, which was low enough to propel WT mice to stay all most 100% time on the 30°C plate, the W137H mice cannot detect the coldness on plate B so that they still spend about 60% of the time on the 5°C plate B. Such a single plate B temperature setting has already clearly demonstrated the deficiency in cold sensing in the W137H mice. We conducted such experiments with multiple plate B temperature values to be more stringent. However, we did not intend to compare the “Time on 30°C” values across all plate B temperature range. Therefore, we chose to use t-test only to compare W137H mice and WT mice under each plate B temperature setting.

Response to Referee #2

We sincerely thank the referee for the positive evaluation and confirmation that all previous concerns were addressed.

We hope that our revisions meet all editorial and reviewer requirements.

We greatly appreciate your time and consideration.

With kind regards,

Fan Yang

Prof. Fan Yang
Zhejiang University
Department of Biophysics
Yuhangtang Road 866, Zhejiang University Zijingang Campus, School of Medicine, Keyan Building B503 room
HangZhou 310058
China

Dear Prof. Yang,

I am very pleased to accept your manuscript for publication in the next available issue of EMBO reports. Thank you for your contribution to our journal.

Yours sincerely,
